

**Gravel threshold of motion: A state function of sediment**
**transport disequilibrium?**
**Joel P. L. Johnson**
Department of Geological Sciences, The University of Texas at Austin
Correspondence to: joelj@jsg.utexas.edu
**Abstract**
In most sediment transport models, a threshold variable dictates the shear stress at which non-
negligible bedload transport begins. Previous work has demonstrated that nondimensional
transport thresholds ($\tau_c^*$) vary with many factors related not only to grain size and shape, but
also with characteristics of the flow and surrounding grains. Both a conceptual model and
flume experiments suggest that $\tau_c^*$ should evolve as a function of local entrainment and
deposition. Net entrainment preferentially removes more mobile particles while leaving
behind more stable grains, gradually increasing $\tau_c^*$ and reducing transport rates. Net
deposition tends to fill in topographic lows, progressively leading to less stable distributions
of surface grains, decreasing $\tau_c^*$ and increasing transport rates. A new model is proposed for
the temporal evolution of $\tau_c^*$ as a power-law function of net erosion or deposition, which
shares some similarities with the Exner equation. Model parameters are calibrated based on
flume experiments that explore transport disequilibrium. The $\tau_c^*$-evolution equation is then
incorporated into a simple morphodynamic model. The evolution of $\tau_c^*$ acts as negative
feedbacks on morphologic change, while also allowing reaches to equilibrate to sediment
supply at different slopes. Finally, $\tau_c^*$ is interpreted to be an important but nonunique state
variable for channel morphology, in a manner consistent with the role that state variables such
as temperature play in describing the evolution of thermodynamic systems.



## 1    Motivation

Despite over a century of quantitative study (Gilbert, 1914), it often remains
challenging to predict coarse bedload transport rates to much better than an order of
magnitude because of the complexity of grain interactions with the flow and the surrounding
grains. Predictive models for complex systems often derive utility from their simplicity, as is
the case with the widely used Meyer-Peter and Müller (1948) transport equation, as modified
by Wong and Parker (2006):
$$q_s^* = 3.97\left(\tau^* - \tau_c^*\right)^{1.5} \qquad \text{for} \qquad \tau^* \geq \tau_c^* \qquad (1)$$
where $q_s^*$ is a nondimensional sediment transport rate per unit width, $\tau^*$ is a nondimensional
shear stress imparted by the fluid on the channel bed (a Shields stress), and $\tau_c^*$ is the
nondimensional threshold stress at which grains begin to move (a critical Shields stress).
Variables are nondimensionalized as followed:
$$q_s^* = \frac{q_s}{D\sqrt{\left(\frac{\rho_s}{\rho}-1\right)gD}} \qquad (2)$$
$$\tau^* = \frac{\tau}{(\rho_s - \rho)gD} \qquad (3)$$
where $q_s$ is volume sediment transport rate per unit width (m²/s), $D$ is grain diameter (m),
$\rho_s$ is sediment density (m³/kg), $\rho$ is water density (m³/kg), $g$ is gravitational acceleration
(m/s²), and $\tau$ is shear stress (Pa).  In principle, these nondimensionalizations should account
for differences in grain size, fluid and sediment density and gravity, allowing meaningful
comparisons of transport and stress across different conditions.  For a given grain diameter
(and ~constant $\rho_s$, $\rho$ and $g$ assumed for terrestrial landscapes), the simplicity of Eq. (1) is
that it predicts transport rate using just two variables, $\tau^*$ (a function of flow strength) and $\tau_c^*$
(a function of many variables). In practice, $\tau_c^*$ is often back-calculated from a particular
sediment transport model.  For example, the Meyer-Peter and Muller (1948) $\tau^*$ and $q_s^*$ give
best-fit $\tau_c^*$=0.0495 for Eq. (1) (Wong and Parker, 2006). In practice, $\tau_c^*$ can essentially be an
empirical fitting parameter for a given transport model (e.g., Wong and Parker, 2006;
Buffington and Montgomery, 1997).



Thresholds of motion for gravel often span an order of magnitude or more (Fig. 1; see
caption for data sources). Variability in $\tau_c^*$ greatly influences bedload flux predictions in
mountain rivers because transport occurs close to thresholds conditions, even during large
floods (Phillips et al., 2013; Parker et al., 1982; Parker and Klingeman, 1982). Previous work
demonstrates a great many factors that can collectively cause $\tau_c^*$ scatter; for example, slope
can empirically explain 34% of the variability shown in Fig. 1 data (e.g., Buffington and
Montgomery, 1997; Kirchner et al., 1990; Lamb et al., 2008). Although interrelated, $\tau_c^*$
influences can generally be classified as grain controls, "bed state" controls, and flow
controls. In addition to diameter and density, grain controls include shape and angularity (e.g.,
Prancevic and Lamb, 2015; Gogus and Defne, 2005). Bed surface controls include the grain
size distribution (GSD) of the surrounding bed, the degree of overlap and interlocking among
grains, the protrusion of grains into the flow, the degree of coarse grain clustering, the bed
roughness, and the reach slope (e.g., Parker, 1990; Wilcock and Crowe, 2003; Sanguinito and
Johnson, 2012; Buscombe and Conley, 2012; Mao, 2012; Kirchner et al., 1990; Strom and
Papanicolaou, 2009; Marquis and Roy, 2012; Powell and Ashworth, 1995; Richards and
Clifford, 1991). Flow characteristics influencing $\tau_c^*$ include particle Reynolds number, flow
depth relative to grain size, the intensity of turbulence, the history of prior flow both above
and below transport thresholds, and the partitioning of stress into form drag and skin friction
(e.g., Shvidchenko and Pender, 2000; Ockelford and Haynes, 2013; Schneider et al., 2015a).
However, most flow-dependent controls are not independent of the bed surface controls. For
example, flow depths, turbulence and form drag depend on slope and bed roughness, while
the stress history influences $\tau_c^*$ by changing grain interlocking and surface roughness.
In addition, recent work suggests that the amount of sediment supplied from upstream
affects $\tau_c^*$, with higher rates of upstream supply corresponding to more mobile sediment and
lower $\tau_c^*$ (Recking, 2012; Bunte et al., 2013). Sensitivity of $\tau_c^*$ to sediment flux is not
obviously classifiable as either a flow or a bed state control. The idea that transport rate
influences $\tau_c^*$ is intriguing because, by definition, $\tau_c^*$ influences transport rate (Eq. 1). This
feedback is the focus of the present analysis. A constant $\tau_c^*$ is commonly assumed for gravel
transport calculations, perhaps for several reasons. First, the traditional Shields diagram
indicates that $\tau_c^*$ is rather insensitive to particle Reynolds number once flow becomes





hydraulically rough around grains (Buffington, 1999).   Second, a belief that $\tau_c^*$ is
fundamentally a material property of a grain rather than a bed state control also remains
somewhat ingrained.   Third, because the best estimate of a given variable is usually its
average, there is a tendency to attribute variability to measurement noise and uncertainty,
even when that variability may be real, its causes understandable, and its influence potentially
important to system dynamics (Jerolmack, 2011; Buffington and Montgomery, 1997).
Feedback    between    channel    morphology    and    bedload    transport    defines
morphodynamics. The Exner equation of sediment mass balance quantifies how transport
changes correspond to topographic changes (Paola and Voller, 2005):
$$\frac{\partial z}{\partial t} = -\left(\frac{1}{1-\lambda_p}\right)\frac{\partial q_s}{\partial x} \qquad (4)$$
where $z$ is bed elevation (vertical position)  $x$ is horizontal position, $t$ is time, and $\lambda_p$ is bed
porosity. Sediment flux and bed elevation are functions of both position and time. In this
morphodynamic equation (presented for simplicity without an uplift or subsidence term),
topographic equilibrium ($\partial z/\partial t = 0$) is attained when the sediment flux into a reach equals the
sediment flux out ($\partial q_s/\partial x = 0$). Channel morphology has long been recognized to influence
sediment transport. Of particular relevance to the present work, Stark and Stark (2001)
proposed an insightful landscape evolution model with a variable called *channelization* that is
defined as representing "the ease with which sediment can flux through a channel reach"
(Stark and Stark, 2001). Conceptually, channelization characterizes how changes in reach
morphology influence local transport rate. However, channelization is an abstract unitless
number that does not correspond physically to any measureable aspects of morphology. A
fundamental feedback is imposed in the model: channelization evolves through time as a
function of both sediment flux and of itself, resulting in a differential equation. The
combination of local slope and channelization tend to asymptote towards values such that
$\partial q_s/\partial x = 0$, i.e. transport equilibrium.  For a given upstream sediment supply rate, a modeled
reach can evolve to equilibrium at different slopes (for different corresponding values of
channelization) because both slope and channelization affect transport rate.
Interestingly, the above definition of channelization could also be applied to $\tau_c^*$.
Because of its control on transport rates, changes in $\tau_c^*$ should influence channel





morphodynamics, both over human timescales (e.g., in response to natural and anthropogenic
perturbations such as landslides, floods, post-wildfire erosion, land use, changing climate) and
longer timescales (landscape evolution). The overall goal of the present work is to elucidate
possible feedbacks among thresholds of motion, changes in transport rate, and the
morphological evolution of channels. A motivating hypothesis is that variability in gravel $\tau_c^*$
is both physically meaningful and reflects changes in $\tau_c^*$ as a predictable function of sediment
flux. Another hypothesis is that the implicit effects of multiple processes on $\tau_c^*$ can
collectively be accounted for in terms of sediment flux dependence. Because changes in
alluvial channel morphology are strongly coupled with sediment flux (Eq. 4), I also
hypothesize that the evolution of $\tau_c^*$ can implicitly model effects of evolving channel
morphology.

The paper is organized as follows. First, a conceptual model for why $\tau_c^*$ should

depend on sediment flux is presented, followed by equations. Next, flume experiments on
mixed grain size transport are used to empirically calibrate model parameters. The
experimental data are consistent with the Wilcock and Crowe (2003) hiding function that
predicts transport rates for grain size mixtures. A simple morphodynamic model is then used
to evaluate how evolving $\tau_c^*$ influences timescales of channel profile evolution. Finally, in
the discussion section $\tau_c^*$ is argued to be one of many "state variables" that can describe how
channels evolve in response to external forcing and internal feedbacks. Comparisons are
made to state variables in thermodynamics.

**2    Models and Methods**
**2.1    Conceptual  framework for $\tau_c^*$ evolution**

For bed surface grains of a given size class, $\tau_c^*$ actually represents the stress at which

only the most mobile of those grains become entrained. Individual grains will each have a
different threshold based on the pocket geometry and near-bed flow velocity at its particular
location, resulting in $\tau_c^*$ probability distributions when considering all similarly sized grains
(Kirchner et al., 1990; Buffington et al., 1992). Gravel flux increases with discharge primarily



because thresholds are gradually exceeded for increasing proportions of surface grains of a
given size. When the sediment flux entering a reach—$q_{\sin}$—is balanced by the flux exiting—
$q_{sout}$—it is assumed that the $\tau_c^*$ probability distribution does not change over time, and that
$\tau_c^*$ remains constant.

In the case of a channel reach undergoing net erosion ($q_{sout} > q_{\sin}$, so $\partial q_s / \partial x > 0$), the

most mobile grains would preferentially be entrained first. Progressive erosion will entrain
grains from increasingly more stable positions on the bed, gradually increasing $\tau_c^*$.
Conversely, during net deposition, grains will tend to preferentially deposit in more stable bed
positions such as local topographic lows. Continued deposition would lead to grains being
deposited in progressively less stable positions, gradually decreasing $\tau_c^*$. These changes
describe negative transport feedbacks: net erosion progressively reduces rates of erosion by
making grains harder to entrain, while net deposition progressively makes grains more mobile
and less likely to be deposited.

The amount by which $\tau_c^*$ changes should also depend on the current state of the bed

surface. If bed surface grains are already loosely packed and highly mobile, additional
deposition would cause little additional decrease in $\tau_c^*$. On the other hand, initial deposition
onto a stable bed would likely cause bigger reductions in $\tau_c^*$ than subsequent deposition.
Thus, the change in $\tau_c^*$ should also be a function of $\tau_c^*$.

Two additional points need to be made. First, natural sediment is a mixture of sizes. It

is common to assume that a single representative grain size, such as the median ($D_{50}$),
adequately describes transport of the whole mixture. Second, "thresholds" can represent
different things in different models. In Eq. (1), $\tau^* = \tau_c^*$ represents a modeled transport rate of
zero. In other models designed to predict measurable transport rates at very low shear stresses,
a non-threshold "reference" stress is instead defined as the Shields stress at which $q_s^*$ has a
very low but specific nonzero value (Parker, 1990; Wilcock and Crowe, 2003). For many
applications the practical difference between threshold and reference stresses are negligible
(Buffington and Montgomery, 1997), and in the present work they are largely used
interchangeably.





## 2.2  $\tau_c^*$-evolution model equations

Because longitudinal coordinate $x$ increases downstream, net deposition in a reach is
indicated by $\partial q_s/\partial x < 0$ and local erosion by $\partial q_s/\partial x > 0$. The following relations are
proposed:
$$\frac{\partial \tau_c^*}{\partial t} = \begin{cases} kB\left(\left|\frac{\partial q_s}{\partial x}\right|\right)^{\kappa_{ent}} & if \quad \partial q_s/\partial x > 0 \\ -k(1-B)\left(\left|\frac{\partial q_s}{\partial x}\right|\right)^{\kappa_{dep}} & if \quad \partial q_s/\partial x < 0 \end{cases} \tag{5}$$

$$B = \frac{\tau_{c\,max}^* - \tau_c^*}{\tau_{c\,max}^* - \tau_{c\,min}^*} \tag{6}$$
where $\kappa_{ent}$ and $\kappa_{dep}$ are dimensionless exponents corresponding to entrainment and
deposition, respectively, and $k$ is a scaling factor. These three parameters will be empirically
fit to experiments. $\tau_{c\,min}^*$ and $\tau_{c\,max}^*$ represent bounds on how low or high $\tau_c^*$ can plausibly
evolve (assumed to be 0.02 and 0.35 respectively). Eq. (5) predicts that $\tau_c^*$ incrementally
decreases when net deposition occurs, and incrementally increases during net erosion in a
reach. "Feedback factor" $B$ makes Eq. (5) a differential equation; it scales the incremental
change in $\tau_c^*$ such that deposition on an already "loose" bed ($\tau_c^*$ close to $\tau_{c\,min}^*$) would be
minimally decrease $\tau_c^*$, but erosion would cause a larger incremental $\tau_c^*$ increase.
Conversely, if $\tau_c^*$ were already high (close to $\tau_{c\,max}^*$), then erosion would cause a much
smaller $\tau_c^*$ change than deposition. Finally, we note that representing $\partial \tau_c^*/\partial t$ as a function of
$\partial q_s/\partial x$ is broadly analogous in form to Exner (Eq. 4).
A limitation of Eq. (5) is that, for dimensionally consistency, the units of $k$ vary with
$\kappa_{ent}$ and $\kappa_{dep}$. An improved equation replaces spatial changes in flux with spatial changes in
the thickness of deposited or eroded sediment:
$$\frac{\partial \tau_c^*}{\partial t} = \begin{cases} kA_rB\left(\left|\frac{\partial \theta_s}{\partial x}\right|\right)^{\kappa_{ent}} & if \quad \partial q_s/\partial x > 0 \\ -kA_r(1-B)\left(\left|\frac{\partial \theta_s}{\partial x}\right|\right)^{\kappa_{dep}} & if \quad \partial q_s/\partial x < 0 \end{cases} \tag{7}$$



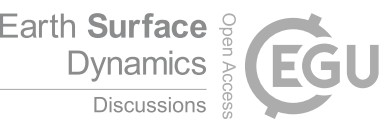

$A_r$ is an optional dimensionless armoring parameter, described further below. $\theta_s$ is the
thickness of sediment deposited or eroded at a given location; it has dimensions of length.
$\partial\theta_s/\partial x$ is a dimensionless ratio representing spatial changes in erosion and deposition. In this
case, $k$ has dimensions $1/t$ and scales how quickly $\tau_c^*$ evolves. $\theta_s$ can calculated by
integrating Eq. (4) over time interval $t_1$ to $t_2$:
$$\theta_s(t_2,x)=z(t_2,x)-z(t_1,x)=\frac{1}{1-\lambda_p}\int_{t_1}^{t_2}\frac{\partial q_s(t,x)}{\partial x}dt \qquad (8)$$
(recall that $\int_a^b (\partial f(s,t)/\partial t)dt = f(s,b)-f(s,a)$ for a generic function $f$). Using discrete flume
data, $\theta_s$ is calculated over a measurement interval $\Delta t$ as $(1-\lambda_p)^{-1}(\overline{q_{sout}}-\overline{q_{sin}})\Delta t/\Delta x$, where
$\Delta x$ is the length of the flume and the sediment flux terms are averaged over $\Delta t$.

While $A_r$ is set to 1 for many calculations below, the parameter is included in order to

explore whether other metrics of relative surface grain size variability or bed roughness
improve predictions. For example, the rate at which $\tau_c^*$ changes might depend on grain size
relative to bed surface roughness ($\sigma$), i.e. $A_r = D_{50}/\sigma$. Setting $A_r = D_{50}/\sigma$ suggests that,
relative to topographic lows and highs, large range cause bigger $\tau_c^*$ changes than small grains.
**2.3   Experimental design**

The flume experiments used to calibrate $k$, $\kappa_{ent}$ and $\kappa_{dep}$ explore how fine gravel

pulses influence the morphodynamics of step-pool like channels. Johnson et al. (2015)
provide details of the experimental conditions and how they scale to natural conditions, and
so the summary here is brief. Four experiments were conducted in a small flume 4 long and
10 cm wide.  Experiments 1 and 4 were done at 8% initial slope, and 2 and 3 at 12% initial
slope.  Water discharge was held constant throughout to better isolate the influence of
sediment supply changes on transport. Sediment transported out of the flume over different
time intervals was caught in a downstream basket, sieved and weighed. Sediment was painted
different colors based on five size classes with median diameters 2.4, 4.5, 8.0, 15.4, and 27.2
mm. Surface GSDs were measured using image analysis of colored bed surface grains during
the experiments. Bed topography was measured using a triangulating laser, and bed roughness
($\sigma$) was calculated from longitudinal topographic swaths as the standard deviation of



detrended bed elevations. Water surface elevations were measured using an ultrasonic
distance sensor, and water depths were calculated by subtracting bed elevations. Total shear
stress ($\tau$) was calculated assuming steady uniform flow when spatially averaged over the
flume:
$\tau = \rho g h S$            (9)
where $h$ is water depth corrected for sidewall effects following the method of Wong and
Parker (2006), and $S$ is water surface slope.
The experiments started with mixed-size sediment screeded flat. Initially, all surface
sizes were observed to be mobile (and therefore above thresholds of grain motion). At the
beginning no sediment was fed into the upstream end ($q_{sin}$=0), and the bed responded by
coarsening, roughening and some net erosion. The bed gradually stabilized and transport rates
dropped to be very low. After this initial bed stabilization, a step-function pulse of the finest
gravel size ($D_{50}$=2.4 mm) was fed into the flume at 1000 g/min, representing an idealization
of a landslide, debris flow, post-wildfire erosion, anthropogenic gravel augmentation, or
similar process that would suddenly supply a prolonged pulse of sediment finer than the
existing bed surface. Initially some deposition occurred on the bed, but the channel adjusted
rapidly, by both entraining coarser bed surface grains and transporting most of the finer
supplied gravel, so that the outlet transport rate ($q_{sout}$) approximately matched $q_{sin}$. After the
sediment supply pulse $q_{sin}$ was again dropped to zero, and the bed gradually restabilized,
asymptotically approaching though not quite attaining equilibrium transport ($q_{sout} \approx 0$) during
the remainder of each experiment.

### 2.4  The Wilcock and Crowe (2003) transport model

As described above, $\tau_c^*$ is often calculated by fitting a sediment transport model to
data. Experimental shear stresses and GSDs provide all of the variables needed to evaluate the
"Surface-based Transport Model for Mixed-Size Sediment" of Wilcock and Crowe (2003),
abbreviated as W&CM. A goal is to evaluate how well the W&CM can predict transport in
the present experiments, which used steeper reach slopes and lower water depths relative to
GSDs compared to experiments of Wilcock and Crowe (2003). In addition, disequilibrium
transport was intentionally quantified during supply perturbations that caused net deposition





or erosion. In contrast, the experiments Wilcock and Crowe (2003) used to calibrate their
model intentionally reflected steady-state transport.

A central variable in the W&CM is $\tau_{rs50}$, the reference stress for the median *surface*

grain size ($D_{s50}$). The nondimensional equivalent is $\tau^*_{rs50}$ (Eq. 3). Rather than being an
actual threshold (i.e., $\tau^*_c$), $\tau^*_{rs50}$ corresponds to a very low transport rate of $W^*_i$ =0.002. $W^*_i$ is
a nondimensional bedload transport rate for grain size class *i*,
$$W^*_i = \left(\frac{\rho_s}{\rho} - 1\right)\frac{gq_{bi}}{F_i u^3_\tau}$$     (10)
where $q_{bi}$ is the volumetric transport rate per unit channel width of grains of size *i*, $F_i$ is the
fraction of size *i* on the bed surface, and $u_\tau$ is shear velocity ($u_\tau = \sqrt{\tau/\rho}$). Wilcock and
Crowe (2003) presented an empirical relationship between transport and shear stress:
$$W^*_i = \begin{cases} 0.002\left(\dfrac{\tau}{\tau_{ri}}\right)^{7.5} & for \quad \tau/\tau_{ri} < 1.35 \\[4mm] 14\left(1 - \dfrac{0.894}{\left(\dfrac{\tau}{\tau_{ri}}\right)^{0.5}}\right)^{4.5} & for \quad \tau/\tau_{ri} \geq 1.35 \end{cases}$$     (11)
where $\tau_{ri}$ is a dimensional reference stress for size class *i*, with dimensionless equivalent $\tau^*_{ri}$
(Eq. 3). A "hiding function" determines how nondimensional reference stresses vary with
grain size:
$$\frac{\tau^*_{ri}}{\tau^*_{rs50}} = \left(\frac{D_i}{D_{s50}}\right)^{b-1}$$     (12)
Hiding function exponent *b* is calculated as
$$b = \frac{0.67}{1 + e^{(1.5 - D_i/D_{s50})}}$$     (13)
Note that the W&CM *b* relation was slightly modified by replacing $D_{sm}$ (the geometric mean
surface diameter) with $D_{s50}$, to more simply use just one measure of the central tendency of




the surface GSD. The additional dependence of $\tau^*_{rs50}$ on surface sand fractions in this model
will be addressed below.

**3    Results**
**3.1    Experimental data and best-fit $\tau^*_{rs50}$, hiding functions**

Transport rates, surface grain size, and reach slopes evolved during the experiments as

the flume beds initially stabilized, then responded to the pulse of sediment supply, and finally
restabilized (Fig. 2). Johnson et al. (2015) explained in detail how the addition of gravels
finer than the stabilized bed surface ultimately caused further surface coarsening. Relevant to
the present analysis, transport rates in and out of the flume are not always balanced (Fig. 2a),
although transport evolves towards this equilibrium condition.

The experimental data are used to determine $W_i^*$ (Eq. 10) and then $\tau^*_{ri}$ for each of the

five size classes (Eq. 11). Best-fit $\tau^*_{rs50}$ is then calculated in two ways. In the first approach, $b$
is simply calculated using Eq. (13). Nonlinear multiple regression (in Matlab) is then used
for best-fit parameter estimation of $\tau^*_{rs50}$ and 95% confidence intervals for each experimental
time step (Fig. 3, "W&C fit"). A key point to emphasize is that experimental grain size
changes do not explain the temporal evolution of best-fit $\tau^*_{rs50}$, because in this analysis the
W&CM should already account for grain size changes in terms of $D_i/D_{s50}$ (Eq. 12, 13). Also
note that a form drag correction was not done; transport was calculated in terms of total stress
(Eq. 9; section 5.5).

While hiding function exponent $b$ varies with relative grain size in the W&CM (Eq.

13), other proposed hiding functions have usually found (or assumed) that a single $b$ value
applies to different grain sizes, at least for a given set of flow and surface conditions (Parker,
1990; Buscombe and Conley, 2012). The second approach explores whether $\tau^*_{rs50}$ estimates
are sensitive to details of the hiding function exponent. Rather than using Eq. (13), $b$ is
empirically fit as single values (for all grain size classes), but with different fits to $b$ for each
separate time step. Nonlinear multiple regression was used to estimate both $b$ and $\tau^*_{rs50}$ in Eq.
(12) (Fig. 3). Although the 95% confidence intervals tend to be larger because both $b$ and





$\tau^*_{rs50}$ were estimated rather than just $\tau^*_{rs50}$, the temporal evolution of experimental $\tau^*_{rs50}$ is
comparable for the two different approaches (Fig. 3).

Interestingly, these experiments are consistent with the particular grain size-dependent

*b* function proposed by Wilcock and Crowe (2003) (Eq. 13). Fig. 4 shows data points
determined using the *b* and $\tau^*_{rs50}$ regressions, *not* using Eq. (13). In spite of substantial scatter
there is a slope break which corresponds to a change in *b* for surface grains smaller and larger
than the median.

### 299    3.2   Calibration of $\tau^*_c$-evolution model parameters

Fig. 5 compares the experiment and W&CM-based calculation of $\tau^*_{rs50}$ to several

predictions of these trends. First, a sand fraction dependence is explored. A unique aspect of
the W&CM not described above is that changes in surface sand fraction could cause temporal
evolution of $\tau^*_{rs50}$. In particular, Wilcock and Crowe (2003) proposed that $\tau^*_{rs50}$ varied
systematically from 0.036 with no surface sand to 0.021 with abundant surface sand:
$$\tau^*_{rs50} = c_1 + c_2 e^{-c_3 F_s} \qquad\qquad\qquad (14)$$
where $F_s$ is the fraction of sand on the bed surface, and constants $c_1$, $c_2$ and $c_3$ were
empirically found by them to be 0.021, 0.015 and 20 respectively (for simplicity, the
geometric mean reference stress was again replaced in their original equation with $\tau^*_{rs50}$).
Subsequent work has shown that thresholds of motion can similarly be reduced by grains
substantially smaller than the bed surface but larger than sand (Venditti et al., 2010; Sklar et
al., 2009; Johnson et al., 2015), suggesting that the "sand fraction" effect could also be
modeled for grains larger than 2 mm.

In the present experiments, the surface fraction of actual sand (<2 mm) was very

small. However, because grains larger than sand but smaller than the average bed surface
may also enhance mobility, we evaluate whether Eq. (14) can explain the experimentally-
constrained variations in $\tau^*_{rs50}$. Using the surface fraction of grains < 2.8 mm (representing the
smallest grain size class in the experiments), a nonlinear multiple regression of Eq. (14) to all
four experiments together yielded a poor although statistically significant fit to the data



($R^2$=0.13; $p$=3x10$^{-5}$; $c_1$=0.097±0.04, $c_2$=0.103±0.11, and $c_3$=5.6±11), confirming that surface
grain size changes alone cannot explain observed $\tau^*_{rs50}$ patterns (Fig. 5, "Sand fraction").
My proposed $\tau^*_c$-evolution models that instead depend on transport disequilibrium
provide much better fits to experimentally-constrained $\tau^*_{rs50}$. Fig. 5 shows model fits using
Eq. (7) with $A_r$=1, both for a single set of model parameters that provide the best fit to all four
experiments together ("collective best fit", $k$=0.17, $\kappa_{dep}$=0.20, $\kappa_{ent}$=0.40), and also best fits
for each of the four experiments separately. The best-fit overall model has $R^2$=0.69,
suggesting statistically that effects of supply and transport disequilibrium can explain over 2/3
of the variability in $\tau^*_{rs50}$ (Table 1). Note that $\tau^*_c$ and $\tau^*_{rs50}$ are assumed to be interchangeable
for these model fits. The procedure used for estimating best-fit model parameters for Eq. (5)
and (7) was different from that described above. The fact that these models are differential
equations prevented the use of nonlinear multiple regressions. Instead, a brute-force approach
was done of incrementally stepping through a wide range of parameter space of $k$, $\kappa_{dep}$ and
$\kappa_{ent}$, and finding the combination of parameters that gave the smallest RMSD. The
calculations of $\tau^*_{rs50}$ through time were started at $\tau^*_{rs50}$=0.036 at $t$=0, which is consistent with
the experiments, and also is the $\tau^*_{rs50}$ proposed by Wilcock and Crowe (2003) in the absence
of sand dependence.
Interestingly, model fits using $A_r = D_{s50}/\sigma$ are not substantially different from $A_r$=1,
and $R^2$=0.69 is the same (Fig. 5). Table 1 includes three additional regressions not shown on
Fig. 5 because the fits overlap almost completely with those already shown.
$A_r = D_{s50}/D_{s84}$ was tried, because the inverse ratio $D_{s84}/D_{s50}$ represents the degree of
armoring in the analysis of $\tau^*_c$ supply dependence by Recking (2012).
$A_r = 2D_{s50}/(D_{s84} - D_{s16})$ was also tried as a measure of the normalized GSD width. Finally,
parameters were estimated for dimensional $\partial q_s/\partial x$ (Eq. 5). Because these variants do not
substantially improve $\tau^*_c$ model fits, we use the simplest dimensionally consistent model (Eq.
7 with $A_r$=1) in the analysis below.





### 346  **3.3  Influence of $\tau_c^*$ evolution on morphodynamics**

### 347  3.3.1 Morphodynamic model development

Next, an idealized morphodynamic model demonstrates how the proposed $\tau_c^*$ relations
influence the evolution of channel reach profiles, focusing on reach slopes and timescales of
adjustment. Because the modeling goal is to isolate and understand effects of evolving $\tau_c^*$,
the underlying model is arguably the simplest reasonable representation of morphodynamic
feedback. Inspired by G. Parker's Morphodynamics e-book, the model describes a channel
reach in which slope evolves through aggradation and degradation. The downstream
boundary elevation is fixed (constant base level). Sediment transport and bed elevation are
modeled using Eq. (1) and (4) with a single grain diameter ($D$). Unit water discharge $q_w$ is
similarly held constant for simplicity. Upstream sediment supply ($q_{sfeed}$) is imposed, and is
varied in simulations below to drive channels to new steady states. Relationships among flow
depth, depth-averaged velocity and discharge are imposed by assuming that hydraulic
roughness remains constant, parameterized though a Darcy-Weisbach hydraulic friction
coefficient:
$$f = \frac{8 g q_w S}{U^3} \qquad (15)$$
For a given discharge this allows both $U$ and $h$ to be determined:
$$U = \frac{q_w}{h} \qquad (16)$$
$$h = q_w^{2/3} \left( \frac{f}{8 g S} \right)^{1/3} \qquad (17)$$
Although not presented, simulations were also done in which the relation between $U$ and $h$
was determined by instead holding Froude number ($Fr = U/\sqrt{gh}$) constant (Grant, 1997).
While $f$ changes systematically with slope in this scenario, resulting trends in reach slope
adjustment and response timescales are substantively the same as shown below for constant $f$,
suggesting little sensitivity to the underlying hydraulic closure assumptions.




Two model variations are compared:  in the "Exner-only" morphodynamic model, $\tau_c^*$
is held constant. In the "Exner+$\tau_c^*$" variant, $\tau_c^*$ evolves through time following Eq. (7).  At
equilibrium, channel slope can be predicted by combining Eq. (1), (2), (9) and (17):
$$S_{eq} = \frac{2.83}{q_w}\left(\frac{g}{f}\right)^{\frac{1}{2}} D^{\frac{3}{2}} \left(\frac{\rho_s}{\rho}-1\right)^{\frac{3}{2}} \left[\left(\frac{q_s^*}{3.97}\right)^{\frac{2}{3}} + \tau_c^*\right]^{\frac{3}{2}}$$    (18)
For a given discharge, Eq. (18) indicates that both sediment supply and the threshold of
motion influence steady-state morphology (slope).
Away from equilibrium, rates of bed elevation change along a river profile should
depend not only on the sediment flux at a given channel cross section, but also on the average
velocity at which grains move downstream. This control has occasionally been ignored in
previous models of profile evolution. In my model, it is crudely incorporated by assuming that
average bedload velocity is a consistent fraction of water velocity, broadly consistent with
previous findings that bedload velocities are proportional to shear velocity (e.g., Martin et al.,
2012). The modeling timestep is set to be equal to the time it takes sediment to move from
one model node (bed location) to the next, and is adjusted during simulations. While this
approach makes the temporal evolution of channel changes internally consistent within the
model, timescales for morphological response will be much shorter than actual adjustment
times in field settings because flood intermittency is not included (so the model as
implemented is always at a constant flood discharge). In addition, the upstream sediment
supply is imposed in the model, while in natural settings hillslope-floodplain-channel
coupling could greatly affect $q_{sfeed}$ over time if significant aggradation or downcutting took
place.
Table 2 provides parameters used for morphodynamic modeling.  Although the highly
simplified model is not intended for quantitative field comparisons, variables $D$ ($D_{s50}$=50
mm), $f$ (0.1), and $q_w$ (1 m$^2$/s) were chosen to be broadly consistent with a moderate ($\approx$2-3 year
peak discharge recurrence interval) bedload-transporting flood in Reynolds Creek, Idaho
(Olinde and Johnson, 2015). Reynolds creek is a snowmelt-dominated channel with reach
slopes that vary widely from ~0.005 to 0.07.  In an instrumented reach with a slope of 0.02,
Olinde (2015) used RFID-tagged tracers and channel-spanning RFID antennas to measure
$\tau_{rs50}^* \approx$0.06.  A constant $\tau_c^*$=0.06 is used for the Exner-only only models, while $\tau_c^*$=0.06 is





used as the initial condition for Exner+$\tau_c^*$ models with evolving $\tau_c^*$. Upstream sediment feed
rates ($q_{sfeed}$) were not constrained by field data, and were chosen to provide reasonable
modeled slopes. Exponents $\kappa_{dep}$ and $\kappa_{ent}$ used the experimental calibrations, while $k$ were
chosen so that changes in $\tau_c^*$ occurred over the same range of timescales as topographic
adjustments, to better illustrate the interplay of variables in morphodynamic evolution.
### 3.3.2 Morphodynamic model results

Fig. 6 compares how the Exner-only (constant $\tau_c^*$) and Exner+$\tau_c^*$ (evolving $\tau_c^*$)

models respond to an increase in sediment supply. The initial condition is a channel at
equilibrium ($q_{sout}=q_{sin}=q_{sfeed}$). At $t$=0, sediment supply is increased by a factor 5. The $\tau_c^*$-
evolution model aggrades to a new equilibrium slope that is lower than the constant $\tau_c^*$ model.
This occurs because $\partial q_s/\partial x < 0$ causes evolving $\tau_c^*$ to decrease over time, progressively
increasing transport efficiency (i.e., higher transport rates at a lower slope) compared to
constant $\tau_c^*$=0.06 (Fig. 7). Feedback causes the reverse effect in response to a decrease in
$q_{sfeed}$ (Fig. 7). For the Exner+$\tau_c^*$ model, $\tau_c^*$ progressively increases as slope decreases,
leading the channel to re-equilibrate both sooner and at a higher slope.

An equilibrium timescale ($t_{eq}$) is measured as the amount of time it takes from a

supply perturbation ($t$=0 in these models) to having the slope adjust to be within 0.0001 of its
asymptotic equilibrium slope. In Fig. 7, $t_{eq}$ are substantially longer for the Exner-only models
(constant $\tau_c^*$) than for the otherwise equivalent Exner+$\tau_c^*$ models. Slope and $\tau_c^*$ adjust at the
same time in the Exner+$\tau_c^*$ models, and influence transport in the same direction. For
example, an increase in $q_{sfeed}$ leads to aggradation, in turn increasing $q_s^*$ by both increasing
slope and decreasing $\tau_c^*$ and (Eq. 1, 5). Both factors adjusting enable equilibrium to be
reached sooner.

Fig. 8a confirms, over a $q_{sfeed}$ range of two orders of magnitude, that equilibrium

slopes changes less for the Exner+$\tau_c^*$ model than for Exner-only. The ratio of these
equilibrium slopes illustrates the magnitude of the change, where "$S_{eq}$ ratio" is $S_{eq}$ for
Exner+$\tau_c^*$ divided by Exner-only $S_{eq}$ (Fig. 8b). An order-of-magnitude decrease in $q_{sfeed}$



caused Exner+$\tau_c^*$ $S_{eq}$ to be between roughly 24% and 36% larger than Exner-only $S_{eq}$, while
an order-of-magnitude increase in $q_{sfeed}$ led to Exner+$\tau_c^*$ roughly 20% smaller than the
constant-$\tau_c^*$ model. Calculations are shown for several values of scaling factor $k$. A larger $k$
means that $\tau_c^*$ increases or decreases more rapidly for a given amount of aggradation or
degradation (Eq. 7), which in general enables a new equilibrium to be reached with a smaller
change in slope.
Equilibrium timescales are quite sensitive to $k$ as well as to sediment supply rate (Fig
8c). Similar to the $S_{eq}$ ratio, the "$t_{eq}$ ratio" is Exner+$\tau_c^*$ $t_{eq}$ divided by Exner-only $t_{eq}$ (Fig. 8d).
There is an asymmetry in equilibrium times for aggradation vs. degradation; in general the
difference between Exner-only and Exner+$\tau_c^*$ is somewhat smaller during bed aggradation,
and the difference decreases with increasing $q_{sfeed}$. Interestingly, the highest $k$ (2.8E-5) results
in a threshold-like response where the $t_{eq}$ ratio rapidly increases from roughly 0.01 to 0.8
(Fig. 8d). This change occurred at the feed rate at which $\tau_c^*$ "bottomed out", i.e. reached its
minimum possible value ($\tau_c^* \approx \tau_{c\,min}^* = 0.02$) before the equilibrium slope had been attained
(Fig. 8e). At that point, $\tau_c^*$ could no longer act as a buffer to reduce more gradual slope
changes.
Finally, Fig. 9 shows that the spatial as well as temporal evolution of $\tau_c^*$ can influence
river profiles. The models are the same as in Fig. 6. At $t$=0, the feed rate *into the upstream-*
*most node* (node 1, 0 km) increases by a factor of 5. Thus the upstream end feels the flux
perturbation both sooner and more strongly than downstream nodes. Aggradation from the
supply perturbation increases upstream slopes first. In the Exner-only model, downstream
slopes gradually catch up (Fig. 9a): because $\tau_c^*$ stays constant, every location along the
channel eventually asymptotes to the single slope required to transport the new $q_{sfeed}$ at the
given discharge. However, for evolving $\tau_c^*$, enhanced upstream aggradation caused upstream
$\tau_c^*$ to decrease both more rapidly and to lower values than downstream nodes. Spatial
differences in $\tau_c^*$ persisted at equilibrium, resulting in spatial variations in equilibrium slope
(Exner+$\tau_c^*$; Fig. 9b, 9c).





**4    Discussion**

In this section, the dependence of $\tau_c^*$ on sediment supply is compared to previous

work. $\tau_c^*$ evolution is identified as a negative feedback on morphologic change that can
impart a memory of previous channel "states" to the system. The physical significance of $\tau_c^*$
variability is interpreted in terms of steady-state slope variability along a natural channel.
Finally, $\tau_c^*$ is interpreted as a channel state variable, analogous to temperature in
thermodynamics. By incorporating $\tau_c^*$ evolution, landscape evolution models may be able to
implicitly account for morphodynamic variables including roughness and form drag.
**4.1    Comparison to previous work**

Recking (2012) compared long-term bed load monitoring records from steep natural

channels (>5% slope) to differences in sediment supply interpreted from aerial photographs of
surrounding hillslopes. Channels with higher supply rates had higher transport rates for a
given shear stress, suggesting a dependence of transport thresholds on supply.  Recking
(2012) proposed models for putting end-member bounds on reference stress for the cases of
very high sediment supply ($\tau_{mss}^*$) and very low sediment supply ($\tau_m^*$) in steep mountain
channels:
$$\tau_{mss}^* = \left(5S + 0.06\right)\left(\frac{D_{84}}{D_{50}}\right)^{-1.5}$$    (19)
$$\tau_m^* = \left(5S + 0.06\right)\left(\frac{D_{84}}{D_{50}}\right)^{4.4\sqrt{S}-1.5}$$    (20)
The ratio $D_{84}/D_{50}$ is intended to represent the degree of armoring, which varies with sediment
supply (e.g., Dietrich et al., 1989).  It should be noted that these reference stresses parameters
were calibrated to describe transport of the $D_{84}$ grain size (rather than $D_{50}$) using a different
transport model (Recking, 2012).  Nonetheless, $\tau_{mss}^*$ and $\tau_m^*$ were shown to be fairly
comparable to typical nondimensional threshold such as $\tau_c^*$ in Eq. (1).  Fig. 10 compares Eq.
(19) and (20) to the experimental constraints on $\tau_{rs50}^*$.  For the most part, $\tau_{mss}^*$ and $\tau_m^*$ do
bound $\tau_{rs50}^*$.  The low supply bound $\tau_m^*$ is roughly 3 times larger than the experimental





constraints.  While  $\tau^*_{mss}$  is similar in magnitude to  $\tau^*_{rs50}$  and predicts the decrease during the
feed period, the (linear) correlation between  $\tau^*_{mss}$  and  $\tau^*_{rs50}$  is weak ($R^2$=0.13) although
statistically significant ($p$=3E-5).  Nonetheless, given that threshold of motion uncertainties
are typically large, Eq. (19) arguably provides a surprisingly good independent prediction of
our experimental disequilibrium transport data, based on experimental slope, $D_{84}$ and $D_{50}$.
Bunte et al. (2013) also interpreted that lower  $\tau^*_c$  corresponded to looser beds caused
by higher rates of sediment supply from upstream, and noted that the stability of bed particles
can be qualitatively assessed in the field while doing pebble counts. Yager et al. (2012b)
demonstrated that in-channel sediment availability varied inversely with the degree of boulder
protrusion. While the  $\tau^*_c$ -evolution model is not inconsistent with high sediment supply rates
correlating with low  $\tau^*_c$ , Eq. (5) and (7) say something different:  $\tau^*_c$  does not directly increase
or decrease with supply, but rather with the history of sediment supply relative to transport
capacity over time. If $q_{sin}$ equals $q_{sout}$,  $\tau^*_c$  will remain constant regardless of whether $q_{sin}$ is
high or low.

### 4.2  Negative feedback and asymmetric approaches to equilibrium

The evolution of  $\tau^*_c$  acts as a negative feedback because it reduces the
morphodynamic response to perturbations. Reach slopes and  $\tau^*_c$  both change in the direction
that brings transport back towards equilibrium, allowing smaller slope changes to accomodate
supply changes (Fig. 6, 7, 8a,b, 9). However, as with other buffered systems, there is a limit to
how large of a perturbation can be accommodated by  $\tau^*_c$  (as illustrated by $k$=2.8E-5 in Fig.
8c,d,e).  The amount of possible  $\tau^*_c$  change depends on how close the  $\tau^*_c$  is to the threshold
boundaries, i.e.  $\tau^*_{c\,min}$  or  $\tau^*_{c\,max}$  (Eq. 5, 7). When changes in  $\tau^*_c$  are no longer possible (or are
asymptotically small) but transport and morphology are not equilibrated, then the time to
equilibrium ($t_{eq}$) increases because only channel morphology can adjust (Fig. 8c, d, e).
The experiments suggest that  $\tau^*_c$  changes faster in response to aggradation than
degradation (Fig. 2, 5). This asymmetry is expressed in the best-fit exponents:  $\kappa_{dep}$  is smaller
than  $\kappa_{ent}$  for all scenarios tested (Table 1).  Note that because  $\partial\theta_s/\partial x$  is much smaller than 1
(i.e, spatial changes in bed elevation are small compared to the horizontal distance the change





is measured over), the smaller exponent ($\kappa_{dep}$) corresponds to a larger change in $\tau_c^*$ for a
given $\partial \theta_s / \partial x$ (Eq. 7). Thus, aggradation is more efficient at decreasing $\tau_c^*$ than degradation
is at increasing $\tau_c^*$, for a given increment of sediment thickness ($\theta_s$). But why then does Fig.
8c indicate, for the Exner+$\tau_c^*$ model, that equilibrium timescales are longer for aggradation
($q_{sfeed}$ / initial $q_{sfeed}$ > 1) than for degradation? The explanation is that the equilibrium
timescale does not *only* depend on the exponents, but also on how much total aggradation or
degradation occurs to attain equilibrium, and how quickly sediment enters or exits the reach
to enable that aggradation or degradation. More slope change occurred during aggradation
than degradation for these particular Exner+$\tau_c^*$ models (Fig. 8a), even though $\tau_c^*$ also tended
to change by more during aggradation than degradation (Fig. 8e).

In the experiments, average slopes changed very little in response to changes in

sediment supply and transport disequilibrium, while grain size and bed surface roughness
changed much more (Fig. 2). Because grain size changes were accounted for (by the W&CM)
in determining experimental $\tau_{rs50}^*$ (Fig. 3), roughness and other unquantified mechanisms
(such as grain interlocking) are interpreted to have caused the $\tau_{rs50}^*$ evolution. What does this
suggest for $k$, which scales how much $\tau_c^*$ changes for a given amount of aggradation or
degradation? The best-fit $k$ to the collective experiments was 2.83E-3 s$^{-1}$, which reflects the
rapid adjustment of $\tau_c^*$ compared to slope changes (Fig. 5, Table 1, Eq 7). In contrast, the
morphodynamic modeling used $k$ values adjusted to be 2 to 3 orders of magnitude smaller, so
that the response to a perturbation in supply would involve non-negligible changes in slope
(the only morphologic variable in the simple morphodynamic model) as well as in $\tau_c^*$. Higher
values of $k$ in the morphodynamic model cause $\tau_c^*$ to adjust more rapidly and slope to adjust
less (Fig. 8).
**4.3   Memory, morphologic variability, and Reynolds Creek**

An implication of $\tau_c^*$ evolving with reach morphodynamics is that local channel form

can retain "memories" of previous conditions, which can influence local responses to
subsequent forcing. In Fig. 9b and 9c, an increase in supply led to the temporal and spatial
evolution of $\tau_c^*$, which in turn caused spatial variations in equilibrium slope. Upstream



reaches acted as filters of the supply perturbation to downstream reaches (here, "reach" refers
to a small fraction of the total modeled longitudinal distance). In contrast, slope variability
developed during the transient adjustment period when $\tau_c^*$ remained constant, but all reaches
evolved to the same equilibrium slope required to transport the new supply (Fig. 9a).

Natural river channels inevitably exhibit morphologic variability at reach scales.  For

example, although the longitudinal profile of Reynolds Creek appears smoothly concave over
a spatial scale of 10 km (Fig. 11a), there is substantial slope variability when calculated for
100 m reaches (Fig. 11b).  This 100 m averaging scale was chosen for the following analysis
because it is sufficiently large to plausibly be used for landscape evolution modeling, while
small enough to capture slope variability along a profile.

How much variability in reach slope could be explained by commonly observed

variability in $\tau_c^*$ (e.g., Fig. 1)?  Eq. (18), which assumes equilibrium transport, can be
rearranged to predict transport thresholds for a given slope, water discharge and
nondimensional sediment flux:
$$\tau_c^* = \frac{(q_w S)^{2/3}}{2D\left(\frac{\rho_s}{\rho}-1\right)}\left(\frac{f}{g}\right)^{1/3} - \left(\frac{q_s^*}{3.97}\right)^{2/3} \qquad (21)$$
Fig. 11c compares a histogram of compiled $\tau_c^*$ (Fig. 1) to a histogram of calculated $\tau_c^*$ using
Eq. (21), based on the reach slopes along Reynolds Creek (Fig. 11b).  Fig. 11d compares
slope predictions calculated using Eq. (18) and the $\tau_c^*$ compilation (Fig. 1) to the 100 m reach
slopes of Reynolds Creek.  Although slope can be influenced by many factors, it appears
plausible that natural slope variability in part reflects variability in thresholds of motion.
While compilations are often assumed to reflect "typical" distributions, there is no reason that
the $\tau_c^*$ values from Fig. 1 should reflect the particular distribution of $\tau_c^*$ for $D_{50}$ in Reynolds
Creek. Beyond a field measurement of 0.06 at a particular location, $\tau_c^*$ distributions are
spatially unconstrained. These calculations also assumed $q_s^*$=2E-3 because it provided a
reasonable overlap of the distributions. Nonetheless, a hypothesis to motivate future work is
that systematic and predictable differences in $\tau_c^*$ would be found for Reynolds Creek reaches
that overlap with the $\tau_c^*$-predicted slope distribution, because slope variations dominantly
reflect adjustments to transport the sediment load from upstream. The steeper Reynolds Creek



slopes are interpreted to reflect additional slope influences, such as the abundance of
boulders. Olinde (2015) did extensive point counts to measure surface GSDs over 11 reaches
along Reynolds Creek, and found that coarse size percentiles (e.g. $D_{84}$) varied greatly but $D_{50}$
had less variation. Larger boulders tend to be immobile, while the more mobile grains (e.g.
$D_{50}$) are transported through all of the reaches (e.g., Yager et al., 2007).

Previous studies have explored the slope dependence of $\tau_c^*$ by mechanistically

explaining mean trends (Prancevic and Lamb, 2015; Lamb et al., 2008) (Fig 1). In addition,
Fig. 3, 9 and 11 suggest that $\tau_c^*$ variability can be both meaningful and physically inherent to
bedload transport, because slope and $\tau_c^*$ inevitably evolve together in response to discharge
and sediment supply. In nature, spatially and temporally-averaged morphodynamic
equilibrium will reflect "channel-forming" discharges and a representative sediment supply
from upstream, but floods, local supply perturbations and history add to local variability in $\tau_c^*$
and morphology.

### 4.4   State function framework for modeling morphodynamics

I argue that $\tau_c^*$ is an important state variable for gravel-bed channels, and outline a

possible state function approach for modeling the morphodynamic evolution of channels. The
term "bed state" has long been informally used to describe collective aspects of local channel
morphology, such as surface GSD and armoring and clustering, that change with relative ease
and influence transport rates (e.g., Church, 2006; Gomez and Church, 1989). Although
explicitly describing $\tau_c^*$ evolution and related river feedbacks in terms of state and path
functions may be novel (to my knowledge), this approach is in many ways conceptually
equivalent to the description by Phillips (2007) of landscape evolution and form in terms of
system states and the importance of history, and similar to other works that describe and
generalize complex channel morphology process and response feedbacks (e.g., Fonstad, 2003;
Phillips, 2011, 2009; Chin and Phillips, 2007; Phillips, 1991; Stark and Stark, 2001; Yanites
and Tucker, 2010).

State variables or state functions are integral to many disciplines, including control

systems engineering and thermodynamics. For example, temperature, pressure, enthalpy and
entropy are some of the many thermodynamic state variables. By definition state variables are
path-independent (Oxtoby et al., 2015). For example, temperature ($T$) describes the amount



of thermal energy per unit of a material. A change in temperature of the material depends on
its initial and final states alone (i.e., $\Delta T = T_2 - T_1$), but does not depend on the path, i.e. the
history of temperatures in between times $t_2$ and $t_1$. In contrast, heat--the flow (transfer) of
thermal energy--is a path function (or process function), not a state variable. Heat flow
between bodies is both controlled by and changes the temperature (the state) of those bodies,
but the amount of total heat transferred does depend on the path. Three other points about
state functions are relevant. First, state functions are often interrelated rather than independent
of one another. For example, Gibbs free energy is state function calculated from temperature,
enthalpy and entropy (Hemond and Fechner, 2014). Second, although traditional state
functions are technically only defined for systems at equilibrium, in practice they are valid
and useful approximations of gradually evolving systems (e.g., Kleidon, 2010). Third, the
evolution of systems involving multiple state variables are usually described with coupled
differential equations.

Channel morphodynamics could be described by a similar framework of state and path

functions. Analogous to heat, the cumulative discharges of both water and sediment are path
functions that drive bed state evolution. Channel morphology can be described by numerous
bed state variables, including but not limited to surface GSD, slope, width, depth, bed
roughness, surface grain clustering, interlocking, overlap and imbrication, and finally $\tau_c^*$.
Analogous to temperature, $\tau_c^*$ is a state variable because, for example, a change in $\tau_c^*$ of 0.01
(from say 0.04 at $t_1$ to 0.05 at $t_2$) does not depend on the progression of values in between.
However, the amount of sediment transported between times $t_1$ and $t_2$ does depend on the
history of $\tau_c^*$, and also influences the history of $\tau_c^*$ (Eq. 5, 7).

It is worth noting that entropy is the state variable perhaps used most often to

characterize channel systems (e.g., Chin and Phillips, 2007; Leopold and Langbein, 1962;
Rodriguez-Iturbe and Rinaldo, 1997). Entropy is often used to provide a closure for
underconstrained sets of equations, by assuming that geomorphic and other systems
inherently maximize their entropy at equilibrium (Kleidon, 2010; Chiu, 1987). A limitation of
some maximum-entropy landscape models is that physically-based surface processes are not
always explicitly modeled, making results difficult to validate and less useful for exploring
landscape responses to perturbations or behavior away from steady state, even if they can





create reasonable-looking equilibrium morphologies (Paik and Kumar, 2010). In contrast,
state function $\tau_c^*$ has a clear process-based meaning.
How could a state function framework improve morphodynamic modeling and
incorporate subgrid-scale channel feedbacks into broader landscape evolution models?
Simplicity is important: the most insight is often gained by using the simplest possible model
that can still capture essential feedbacks over spatial and temporal scales of interest. As
described in the introduction, previous work has shown that $\tau_c^*$ is influenced by many
characteristics of the surrounding bed surface. In other words, $\tau_c^*$ is closely related to other
bed state variables. It would probably be impractical to develop or apply a "complete" $\tau_c^*$
model that explicitly incorporated separate state variables for multiple known controls on $\tau_c^*$.
Instead, the $\tau_c^*$ evolution equation (Eq. 5, 7) attempts to strike a balance between predicting
process in a physically justifiable (but empirically calibrated) way, while remaining broadly
applicable. It could similarly be impractical to apply a "complete" reach-scale
morphodynamic model that explicitly parameterized myriad feedbacks using every
corresponding bed state variable. Instead, because many bed state variables are not actually
independent of $\tau_c^*$, aspects of morphology can be subsumed into evolving $\tau_c^*$ for modeling
purposes. This is similar to the channelization approach of Stark and Stark (2001). For
example, in the experiments the bed responded to transport disequilibrium primarily by
changing roughness but not slope. However, roughness was not an explicit, separate variable
in the best-fit $\tau_{rs50}^*$ calculations. Instead, some effects of evolving roughness and other bed
state controls (imbrication, clustering) on transport rates became implicitly accounted for in
the experimentally calibrated $\tau_{rs50}^*$ ($\approx \tau_c^*$). The best-fit model parameters ($k$, $\kappa_{dep}$, $\kappa_{ent}$; Eq. 7;
Table 1) would presumably change if a separate differential equation was developed to
explicitly describe another state variable that is currently implicit in $\tau_c^*$, such as bed
roughness. It is also important to note that bed roughness could have other effects, such as
modifying hydraulic roughness, that are not implicitly captured by $\tau_{rs50}^*$ (Schneider et al.,
2015b).



### 4.5 Form drag vs. parsimony


Calculations of best-fit $\tau^*_{rs50}$ and transport rates used total shear stress (Eq. 9), rather

than partitioning stress into form drag and a lower effective stress used for calculating
transport rates (skin friction). Although not a state variable, form drag is physically justifiable
because larger clasts that protrude higher into the flow (e.g. stable boulders) tend to account
for a disproportionate amount of the total stress through drag, turbulence generation and
pressure gradients. Form drag corrections have been incorporated into many transport models
to enable reasonable transport rates to be calculated using $\tau^*_c$ values typical of systems
without form drag (e.g., Rickenmann and Recking, 2011; David et al., 2011; Yager et al.,
2012a). Conversely, another common approach is simply to use higher $\tau^*_c$ (e.g., Bunte et al.,
2013; Lenzi et al., 2006), consistent with acknowledging that $\tau^*_c$ can be a physically
meaningful fitting parameter to predict transport. Using field data on steep channel gravel
transport, Schneider et al. (2015a) recently compared transport predictions based on (a) form
drag corrections and (b) higher reference stresses. For the most part they found that either
approach could provide similar accuracy. They also noted that "uncertainties in predicted
transport rates remain huge (up to roughly 3 orders of magnitude)" (Schneider et al., 2015a),
and suggested that factors including supply effects may account for remaining discrepancies.
Although beyond the scope of the present analysis, form drag effects could be separated from
best-fit $\tau^*_{rs50}$ by using a calculated skin friction stress rather than total stress. However, doing
so would add extra uncertainty to the shear stresses, while still not accounting for effects of
sediment supply. Implicitly subsuming form drag into $\tau^*_c$ arguably provides a simpler and
more parsimonious approach for modeling transport and morphodynamics.

### 5 Conclusions


Flume data and a corresponding model demonstrate that nondimensional critical shear
stress ($\tau^*_c$) evolves through time as a function of sediment transport disequilibrium. Net
erosion tends to increase local $\tau^*_c$ (reducing transport rates), while net deposition tends to
decrease $\tau^*_c$ (increasing transport rates). This $\tau^*_c$ dependence on sediment supply relative to
transport capacity can plausibly explain much of the ~order-of-magnitude variability almost
inevitably observed in transport threshold data (Fig. 1). This view contrasts with a more




conventional approach in which thresholds of motion are assumed to remain constant in time,
and $\tau_c^*$ variability is treated as noise rather than interpretable signal.
Flume experiments measured sediment transport and channel response away from
equilibrium, as beds stabilized and responded to pulses of gravel.  Because studies of
transport disequilibrium are relatively uncommon, much of our ability to predict transport
rates implicitly assumes steady state conditions.  The experiments explore a parameter space
typical of step-pool morphologies, with steep slopes (8-12%), flow depths comparable to the
coarsest grain size fractions, and mixed grain size transport (Johnson et al., 2015). While $\tau_c^*$
has physical meaning, it is also a fitting parameter that depends on the choice of sediment
transport model. The experimental data are fit by the Wilcock and Crowe (2003) model for
mixed grain size transport, to back-calculate both hiding function exponents and transport
thresholds (which in this model are nondimensional reference stresses, $\tau_{rs50}^*$ ). The Wilcock
and Crowe (2003) hiding function provides a good prediction of our data (including different
exponents for relatively small and large grains), supporting its applicability to steep channels.
A new differential equation is proposed for predicting the temporal evolution of $\tau_c^*$ as a
function of $\partial q_s / \partial x$ (Eq. 5-7). After empirically fitting three model parameters, this $\tau_c^*$-
evolution equation can explain nearly 70% of the variability in experimental $\tau_c^*$. We
incorporate the $\tau_c^*$-evolution equation into a simple model for channel slope evolution.
Changes in $\tau_c^*$ are negative feedbacks on morphodynamic response, because not only slope
but also $\tau_c^*$ evolve when perturbed.  Evolving $\tau_c^*$ also reduces response timescales, allowing
channels to more rapidly adjust to new conditions.  In addition, spatial and temporal
variations in sediment supply can lead to reaches becoming equilibrated over a range of
slopes, because $\tau_c^*$ also influences the equilibrium slope.
Finally, $\tau_c^*$ is interpreted to be a state variable for fluvial channels.  State functions and
path functions are fundamental to many disciplines such as thermodynamics, because they
allow the evolution of systems to be calculated. The same should be true for channels;
sediment transport is to heat as $\tau_c^*$ is to temperature. More broadly, conceptualizing
morphodynamic and landscape evolution models in terms feedbacks among evolving state



variables and path functions may improve our ability to predict landscape responses to land
use, climate change and tectonic forcing.

**Acknowledgements**
I thank A. Aronovitz for conducting the flume experiments, W. Kim for aiding in the
experimental design, L. Olinde for Reynolds Creek analysis and helpful discussions, and M.
Lamb and J. Prancevic for sharing their $\tau_c^*$ data compilation.  I also thank the kind-hearted
reviewers and editor.  Support came from NSF grant EAR-1053508.

**Appendix 1   List of variables**
$A_r$        Dimensionless parameter for incorporating grain size or roughness ratios in

Eq. (7) [1]

$b$          Dimensionless hiding function exponent; either described by Eq. 13 or fit as a

single value [1]

$B$          Dimensionless "feedback factor"; Eq. 6 [1]
$c_1, c_2, c_3$   Dimensionless empirical constants in Eq. (14) [1]
$D$          Grain diameter, for model cases with a single size only [L]
$D_{50}$       Median grain diameter [L]
$D_{s50}$      Median grain diameter of bed surface [L]
$D_i$        Grain diameter of size class $i$ [L]
$f$          Darcy-Weisbach hydraulic friction coefficient; Eq. (15) [1]
$Fr$         Froude number [1]
$F_i$        Areal fraction of grain size class $i$ on the bed surface; Eq. 10 [1]
$F_s$        Areal fraction of sand on the bed surface; Eq. 14 [1]
$g$          Gravitational acceleration [$LT^{-2}$]
$h$          Water depth [L]



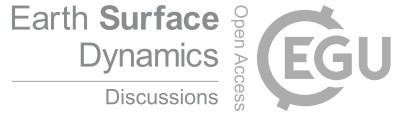

| 733 | $\kappa_{dep}$ | Exponent for net deposition in $\tau_c^*$-evolution models; Eq. (5), (7). [1] |
| 734 | $\kappa_{ent}$ | Exponent for net erosion in $\tau_c^*$-evolution models; Eq. (5), (7). [1] |
| 735 | $k$ | Scaling factor for $\tau_c^*$ evolution. Dimensions are [1/T] for Eq. (7) |
| 736 | $\lambda_p$ | Bed porosity [1] |
| 737 738 | $q_{bi}$ | Volume sediment flux per unit width of size class $i$ in Wilcock and Crowe (2003); Eq. 10 [$L^2$/T] |
| 739 | $q_s$ | Volume sediment flux per unit width [$L^2$/T] |
| 740 | $q_s^*$ | Nondimensional volume sediment flux; Eq. (1)      [1] |
| 741 | $q_{sin}$ | Sediment flux entering a channel bed area (reach) of interest [$L^2$/T] |
| 742 | $q_{sout}$ | Sediment flux exiting a channel bed area (reach) of interest [$L^2$/T] |
| 743 | $q_{sfeed}$ | Sediment flux entering upstream end of overall model domain [$L^2$/T] |
| 744 | $q_w$ | Volume water discharge per unit width [$L^2$/T] |
| 745 | $\rho$ | Water density [M/$L^3$] |
| 746 | $\rho_s$ | Sediment density [M/$L^3$] |
| 747 | $S$ | Water surface slope [1] |
| 748 | $S_{eq}$ | Water surface slope when reach is at equilibrium [1] |
| 749 750 | $\sigma$ | Bed roughness, measured here as the standard deviation of detrended bed elevations [L] |
| 751 | $\theta_s$ | Thickness of sediment deposited or eroded in a time interval; Eq. (7) [L] |
| 752 | $t$ | Time [T] |
| 753 | $t_{eq}$ | Equilibrium timescale for morphological adjustment [T] |
| 754 | $\tau$ | Shear stress [$MT^{-2}L^{-1}$] |
| 755 | $\tau^*$ | Shields stress (nondimensional shear stress) [1] |





| 756 | $\tau_c^*$ | Critical Shields stress (nondimensional critical shear stress) [1] |
|---|---|---|
| 757 | $\tau_{c\,max}^*$ | Imposed maximum bound for $\tau_c^*$ in Eq. (5), (7) [1] |
| 758 | $\tau_{c\,min}^*$ | Imposed minimum bound for $\tau_c^*$ in Eq. (5), (7) [1] |
| 759<br>760 | $\tau_{mss}^*$ | High sediment supply nondimensional reference stress end-member bound in Recking (2012) transport model; Eq. (19) [1] |
| 761<br>762 | $\tau_m^*$ | Low sediment supply nondimensional reference stress end-member bound in (Recking, 2012) transport model; Eq. (20) [1] |
| 763 | $\tau_{ri}^*$ | Reference Shields stress for size class $i$, from Wilcock and Crowe (2003) [1] |
| 764 | $\tau_{rs50}^*$ | Nondimensional reference Shields stress for surface grains of size $D_{s50}$ [1] |
| 765 | $U$ | Depth-averaged water velocity [L] |
| 766 | $u_\tau$ | Shear velocity; Eq. (10) [L/T] |
| 767 | $x$ | Position measured horizontally (distance along channel) [L] |
| 768 | $z$ | Position measured vertically (bed elevation)[L] |
| 769<br>770 | $W_i^*$ | Nondimensional bedload transport rate for grain size class $i$, in Wilcock and Crowe (2003) [1] |
| 771 | W&CM | Abbreviation for Wilcock and Crowe (2003) transport model. |


**Captions**
Figure 1. Threshold of motion data from both field and experimental studies. A power law
regression to these data gives $R^2$=0.34, indicating that a majority of the variability is not
explained by slope alone. Dotted lines indicate common range of $\tau_c^*$=0.03 to 0.06 often
assumed for modeling transport, although measured data fall well out of this range. Data
have been additionally filtered to only include $D_{50} > 2$ mm (i.e. gravel) and slopes between
0.002 and 0.2. Data were compiled and provided by Prancevic and Lamb (2015), based in
part on Buffington and Montgomery (1997), with additional data from Olinde (2015) and
Lenzi et al. (2006).




Figure 2. Flume experiment data (Johnson et al., 2015). a. Sediment transport rate in ($Q_{sfeed}$)
and out of the flume. The upstream sediment supply rate was zero other than during the $Q_{sfeed}$
period. Experiment 1 was run for a longer duration than the others but shows similar trends.
Note that the outlet $Q_s$ adjusts much faster to match the increase in supply than it does to
decrease during periods of no input. b. Median bed surface grain diameters decreased during
the feed of finer gravel, and then increase beyond their previous stable bed. c. Flume-
averaged bed slopes changed relatively little even as transport rates and $D_{50}$ changed greatly
in response to initial bed stabilizing and supply perturbations.

Figure 3. $\tau^*_{rs50}$ fits to the experimental data with the W&CM. ,"W&C fit" uses Eq. (13) to
calculate hiding function exponent b, while "Power-law fit" calculates a best-fit b along with
$\tau^*_{rs50}$. Error bars give 95% confidence intervals on $\tau^*_{rs50}$ based on the regressions; although
uncertainty can be broad the trends are clear and consistent. Shaded area indicates times of
fine gravel addition (sediment feed) in each experiment.

Figure 4. Data points are based on power-law fits to exponent b. The W&CM hiding function
(Eq. 13) does a good job matching the data, although it was not fit to these points. The first 6
measurements of each experiment (roughly the first 10 minutes) were excluded because of
large scatter associated with the greatest bed instability. The plot axes reflect the left and
right hand sides of Eq. (12), although the plot uses dimensional stresses to be consistent with
plots shown in Wilcock and Crowe (2003).

Figure 5. Best-fit models (Eq. 7 and 14) compared to experimental constraints. The periods
of upstream sediment supply ($Q_{sfeed}$) are indicated by the grey boxes for each experiment.

Figure 6. Profile evolution, comparing the morphodynamic responses of models with and
without threshold evolution. The initial condition is an equilibrium channel with $\tau^*_c$=0.06,
upstream sediment supply $q_s$=1e-3 m$^2$/s, and an initial equilibrium slope of 0.0147. Sediment



supply is increased 5x at t=0. Lines are each 5 model days apart, and indicate the evolution to
a new transport equilibrium.

Figure 7. Slope and critical shear stress evolution, for sediment supply increases (which
correspond to Fig. 6 models) and decreases by factors of 5. As in figure 6, t=0 corresponds to
an equilibrium condition where the initial slope and initial threshold are consistent with the
initial upstream sediment supply. Slope and $\tau_c^*$ were averaged over nodes 3-10, leaving out
the first and last two nodes because of minor model boundary effects.

Figure 8. Morphodynamic model sensitivity to sediment supply perturbations and $k$. All
models started at the same equilibrium condition as shown in Fig. 6 and 7. a. Slope
adjustment, normalized by the initial equilibrium slope. The correspondence of Eq. 17 and the
morphodynamic model calculations demonstrate that the models did asymptotically attain
equilibrium slopes.    b. $S_{eq}$ ratio is the ratio of equilibrium slopes of the Exner+$\tau_c^*$ model
divided by $S_{eq}$ for the Exner-only model, to show the relative affect that that $\tau_c^*$ evolution has
on equilibrium slopes. c. Equilibrium timescales for model adjustment. d. $t_{eq}$ ratio is the ratio
of $t_{eq}$ for the Exner+$\tau_c^*$ model divided by $t_{eq}$ for the Exner-only model. Values are lower than
1, indicating that the $\tau_c^*$ evolution has a large influence on equilibrium timescales. e.
Evolution of $\tau_c^*$.

Figure 9. Spatial and temporal evolution of morphodynamic slopes, for the same models
shown in Fig. 6. Slope is initially at equilibrium and responds to the 5x increase in upstream
sediment supply at t=0. a. The Exner-only model initially has spatial slope variability, but
evolves to a uniform new equilibrium slope. b, c. In the model with evolving $\tau_c^*$, slope and
$\tau_c^*$ variability persist even at equilibrium.





Figure 10. Comparison of experimental and best-fit model constraints on $\tau_{rs50}^{*}$, compared to
proposed constraints for D84 reference stress bounds for low and high sediment supply from
Recking (2012).

Figure 11. Comparison of Reynolds creek and $\tau_{c}^{*}$ data compilation predictions using Eq. (18)
and (21). a. Longitudinal profile based on airborne Lidar (Olinde, 2015). Upstream end
(distance=0) is an arbitrary location along the channel (a bridge). Data gap at 730 m is a
gauging station weir; the slope steepens downstream of the weir where the valley becomes
constricted by bedrock, although the bed remains almost entirely alluvial. b. Slopes
calculated (averaged) over 100 m reaches, illustrating reach-scale slope variability. c.
Histogram of $\tau_{c}^{*}$ compilation (data shown in Fig. 1), compared to $\tau_{c}^{*}$ calculated using Eq.
(21), based on Reynolds Creek 100 m slopes (panel b), and assuming $q_w$=1 m2/s, $q_{s}^{*}$=2e-3,
and $f$=0.1. d. Similar calculation as panel c, but using Eq. 18 to solve for slope as a function
of $\tau_{c}^{*}$ from the Fig. 1 data compilation, and compared to panel b slopes.







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





Table 1. Best-fit threshold evolution models

| Model type | | Best-fit coefficients | | | |
| --- | --- | --- | --- | --- | --- |
| | | $c_1$ | $c_2$ | $c_3$ | $R^2$ |
| "Sand fraction" $F_i$ <2.8 mm, collective best fit | Eq. 14 | 0.097 (0.057, 0.14)[a] | 0.103 (0.009, 0.22)[a] | 5.47 (-16.8, 5.8)[a] | 0.13 |
| | | $k$ | $\kappa_{dep}$ | $\kappa_{ent}$ | $R^2$ |
| Units | | 1/s | 1 | 1 | |
| $\partial\theta_s/\partial x$ model, collective fit | Eq. 7, $A_r$ =1 | 2.83E-03 | 0.2 | 0.4 | 0.69 |
| $\partial\theta_s/\partial x$ model, Expt 1 fit | Eq. 7, $A_r$ =1 | 4.12E-02 | 0.55 | 0.61 | 0.52 |
| $\partial\theta_s/\partial x$ model, Expt 2 fit | Eq. 7, $A_r$ =1 | 4.80E-02 | 0.55 | 0.73 | 0.73 |
| $\partial\theta_s/\partial x$ model, Expt 3 fit | Eq. 7, $A_r$ =1 | 2.83E-03 | 0.22 | 0.41 | 0.77 |
| $\partial\theta_s/\partial x$ model, Expt 4 fit | Eq. 7, $A_r$ =1 | 2.75E-02 | 0.43 | 0.62 | 0.75 |
| $\partial q_s/\partial x$ model, collective fit | Eq. 5 | 9.83E-03[b] | 0.25 | 0.4 | 0.69 |
| $\partial\theta_s/\partial x$ model, collective fit | Eq. 7, $A_r$ =$D_{s50}/\sigma$ | 4.17E-03 | 0.24 | 0.43 | 0.69 |
| $\partial\theta_s/\partial x$ model, collectivet fit | Eq. 7, $A_r$ =$D_{s50}/D_{s84}$ | 4.83E-03 | 0.23 | 0.41 | 0.69 |
| $\partial\theta_s/\partial x$ model, collective fit | Eq. 7, $A_r$ =$2D_{s50}/(D_{s84}-D_{s16})$ | 3.83E-03 | 0.23 | 0.42 | 0.69 |

[a]Confidence intervals are +-95%, based on nonlinear multiple regression in Matlab

[b]Units on k for Eq. 5 vary with kent and kdep



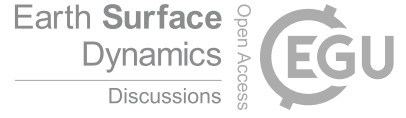

Table 2. Morphodynamic Model Parameters
Constant independent variables:

|  |  | Units |
| --- | --- | --- |
| $q_w$ | 1.00 | m2/s |
| $D_{50}$ | 50 | mm |
| f | 0.1 | |
| h | 0.50 | m |
| U | 1.99 | m/s |
| $\lambda_p$ | 0.25 | |
| $\kappa_{dep}$ | 0.2 | |
| $\kappa_{ent}$ | 0.4 | |
| $\tau^*_{cmin}$ | 0.02 | |
| $\tau^*_{cmax}$ | 0.35 | |
| $\rho$ | 1000 | kg/m$^3$ |
| $\rho_s$ | 2600 | kg/m$^3$ |
| Total duration | 0.5 | years |
| # nodes | 12 | |
| node spacing | 100 | m |
| $U_{bedload}/U$ | 0.5 | |

Initial condition:

| Initial condition: |  | Units |
| --- | --- | --- |
| $q_{sin}$, init | 1.00E-03 | m$^2$/s |
| S, init | 0.0147 | |
| $\tau_{cr}^*$, init | 0.06 | |

Variables changed:

| Variables changed: |  | |
| --- | --- | --- |
| $q_{sin}$ | 1E-5 to 1E-1 | m$^2$/s |
| k | 2.8E-6, 5.7E-6, 2.8E-5 | 1/s |

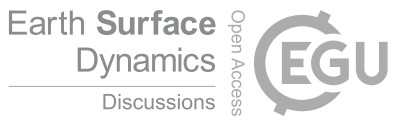

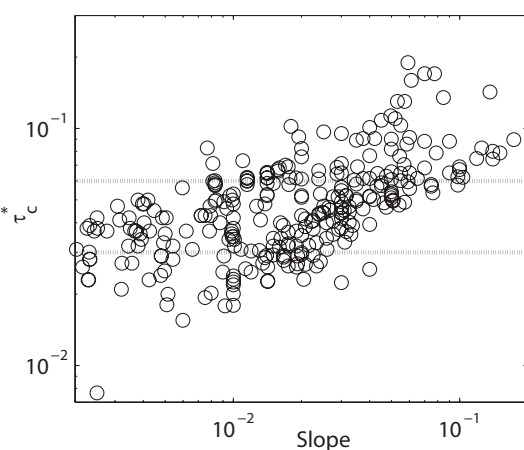

Figure 1. Threshold of motion data from both field and experimental studies. A power law regression to these data gives R²=0.34, indicating that a majority of the variability is not explained by slope alone. Dotted lines indicate common range of 0.03 to 0.06 often assumed for modeling transport, although measured data fall well out of this range. Data have been additionally filtered to only include D50 > 2 mm (i.e. gravel) and slopes between 0.002 and 0.2. Data were compiled and provided by Prancevic and Lamb (2015), based in part on Buffington and Montgomery (1997), with additional data from Olinde (2015) and Lenzi et al. (2006).





Figure 2. Flume experiment data (Johnson et al., 2015). a. Sediment transport rate in ($Q_{sfeed}$) and out of the flume. The upstream sediment supply rate was zero other than during the $Q_{sfeed}$ period. Experiment 1 was run for a longer duration than the others but shows similar trends. Note that the outlet $Q_s$ adjusts much faster to match the increase in supply than it does to decrease during periods of no input. b. Median bed surface grain diameters decreased during the feed of finer gravel, and then increase beyond their previous stable bed. c. Flume-averaged bed slopes changed relatively little even as transport rates and $D_{50}$ changed greatly in response to initial bed stabilizing and supply perturbations.





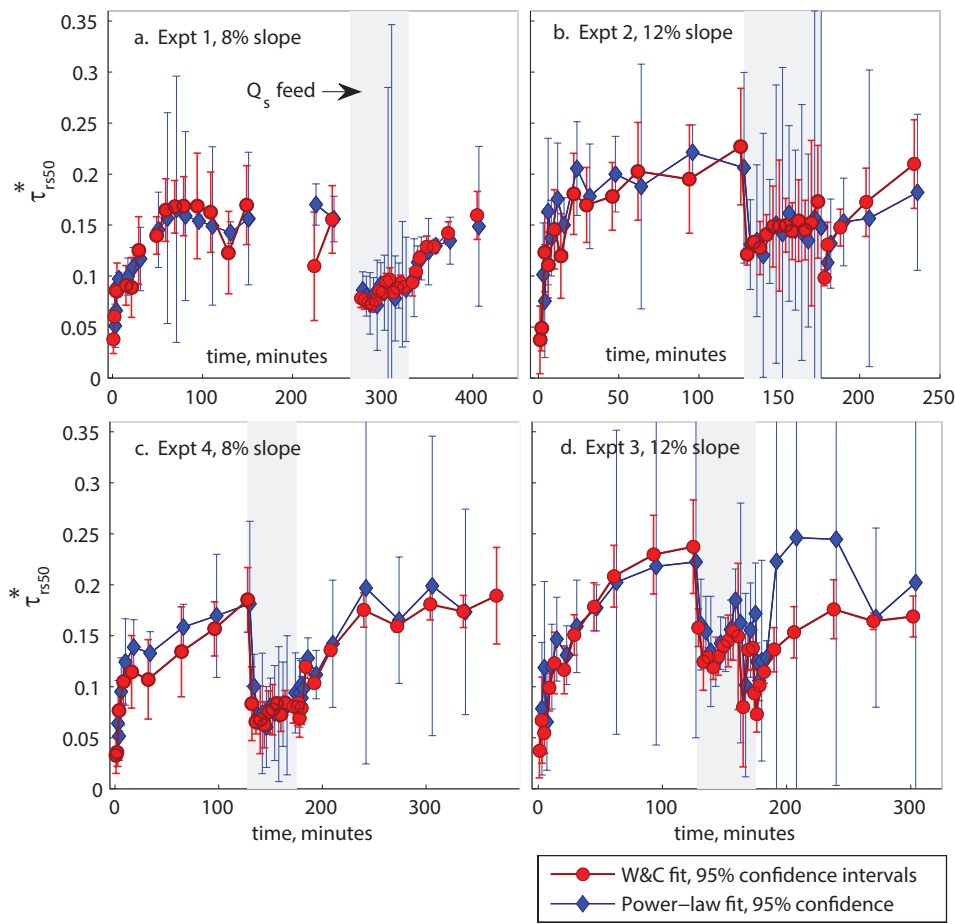

Figure 3. Fits to the experimental data with the W&CM. ,"W&C fit" uses Eq. (13) to calculate hiding function exponent b, while "Power-law fit" calculates a best-fit b along with the threshold parameter. Error bars give 95% confidence intervals based on the regressions; although uncertainty can be broad the trends are clear and consistent. Shaded area indicates times of fine gravel addition (sediment feed) in each experiment.





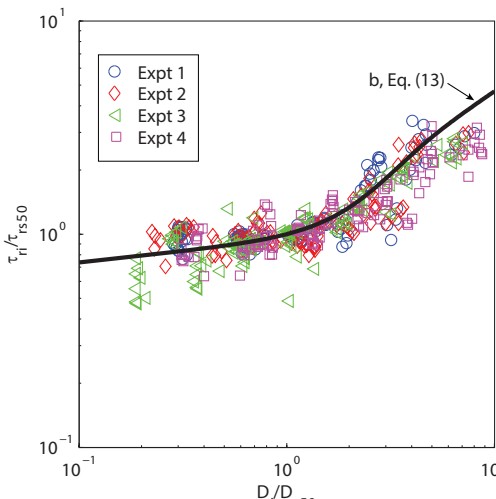

Figure 4. Data points are based on the "Power-law" fits for exponent $b$. The W&CM hiding function (Eq. 13) does a good job matchign the data, although it was not fit to these points. The first 6 measurements of each experiment (roughly the first 10 minutes) were excluded because of large scatter associated with the greatest bed instability. The plot axes reflect the left and right hand sides of Eq. (12), although the plot uses dimensional stresses to be consistent with plots shown in Wilcock and Crowe (2003).




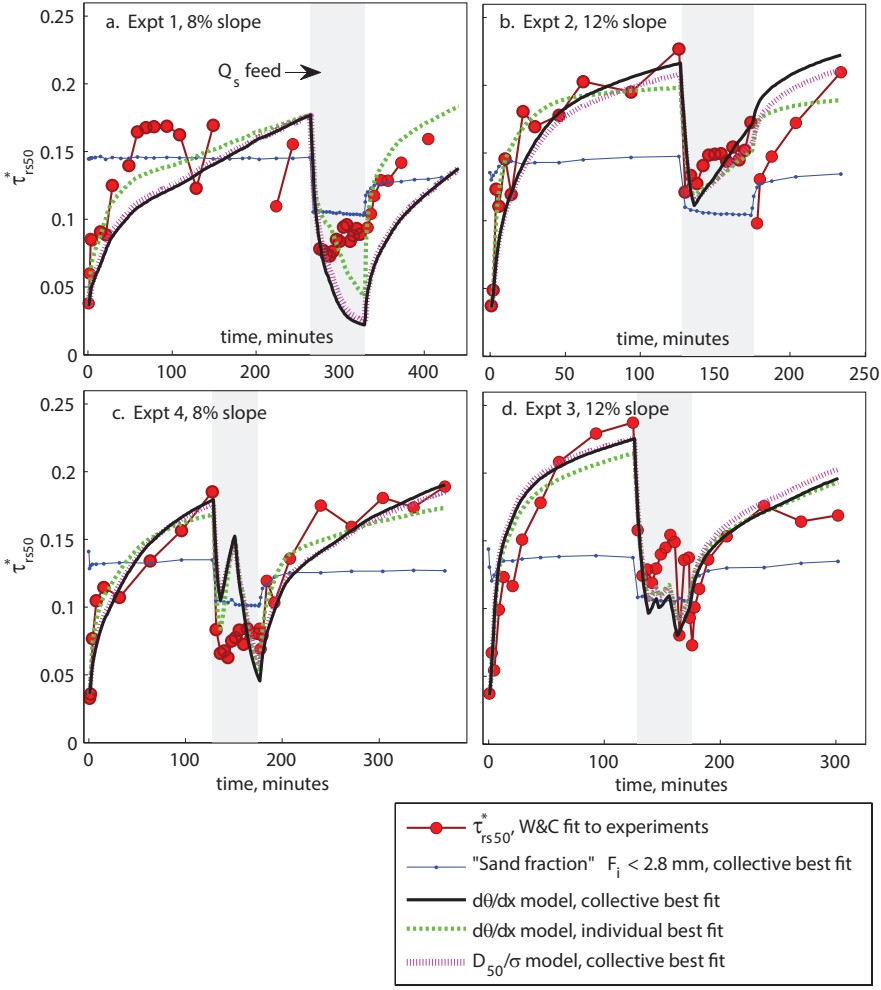

Figure 5. Best-fit models (Eq. 7 and 14) compared to experimental constraints. The periods of upstream sediment supply ($Q_{sfeed}$) are indicated by the grey boxes for each experiment.




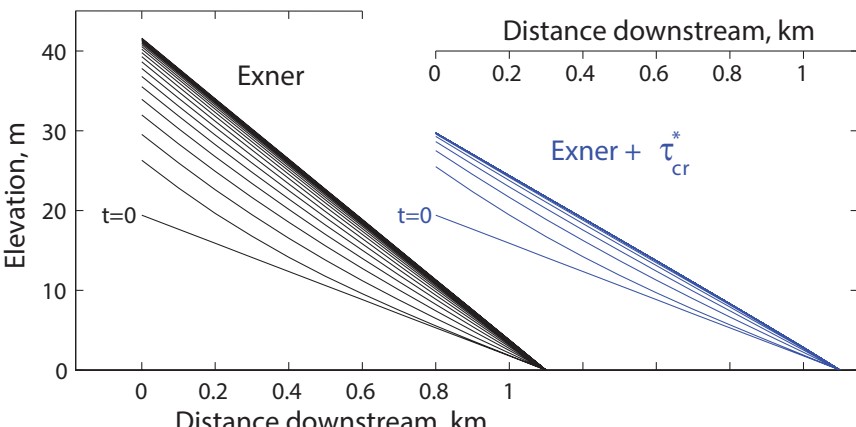

Figure 6.  Profile evolution, comparing the morphodynamic responses of models with and without threshold evolution.  The initial condition is an equilibrium channel with $\tau_c^*$=0.06, upstream sediment supply $q_s$=1e-3 m²/s, and an initial equilibrium slope of 0.0147.  Sediment supply is increased 5x at t=0.  Lines are each 5 model days apart, and indicate the evolution to a new transport equilibrium.





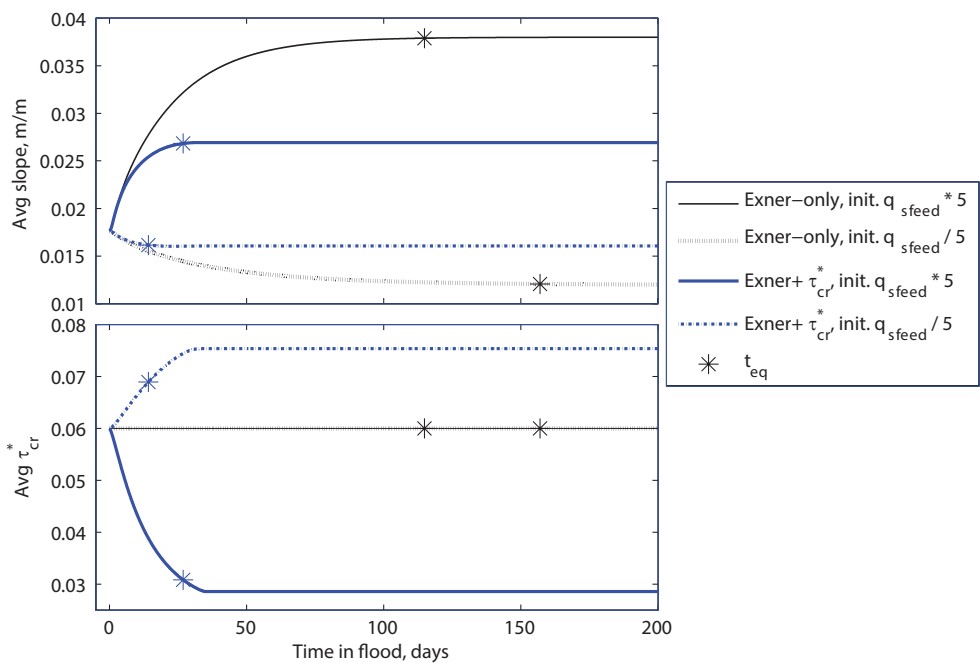

Figure 7. Slope and critical shear stress evolution, for sediment supply increases (which correspond to Fig. 6 models) and decreases by factors of 5. As in figure 6, t=0 corresponds to an equilibrium condition where the initial slope and initial threshold are consistent with the initial upstream sediment supply. Slope and $\tau_c^*$ were averaged over model nodes 3-10, leaving out the first and last two nodes because of minor model boundary effects.



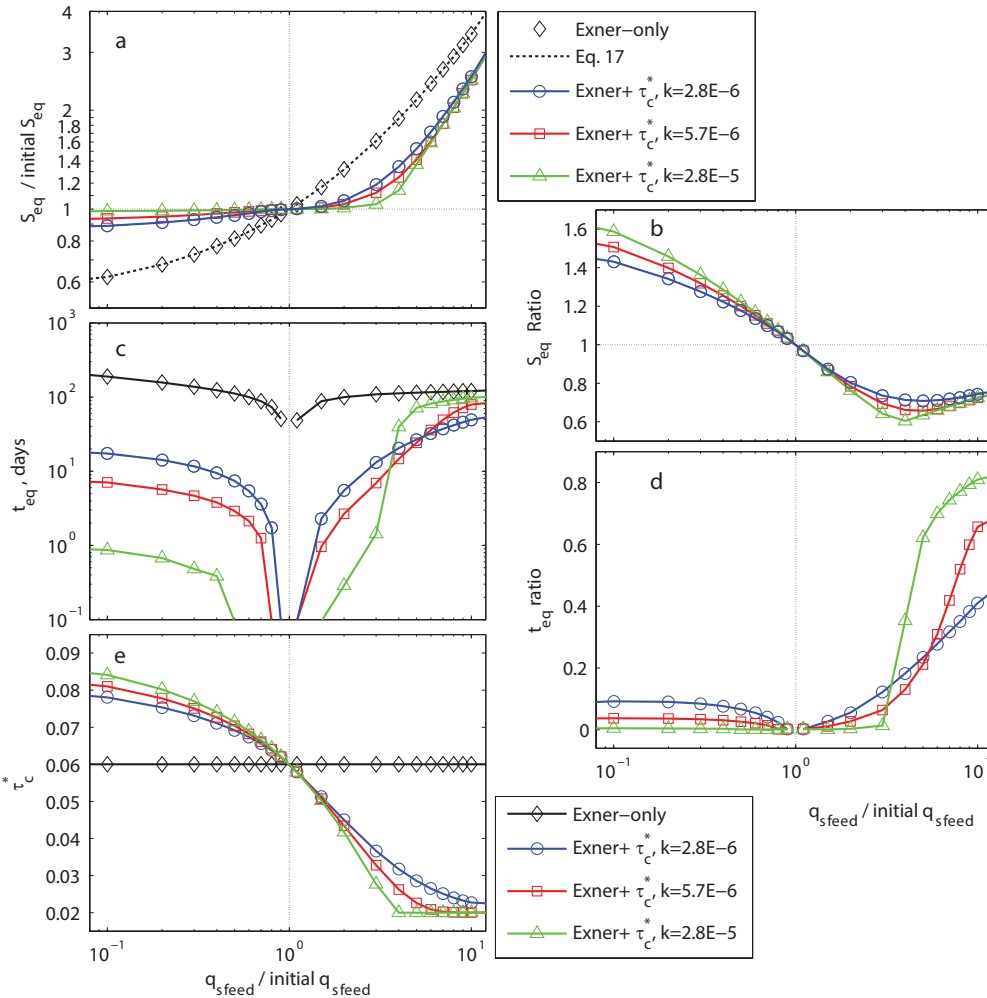

Figure 8. Morphodynamic model sensitivity to sediment supply perturbations and k. All models started at the same equilibrium condition as shown in Fig. 6 and 7. a. Slope adjustment, normalized by the initial equilibrium slope. The correspondence of Eq. 17 and the morphodynamic model calculations demonstrate that the models did asymptotically attain equilibrium slopes. b. $S_{eq}$ ratio is the ratio of equilibrium slopes of the Exner+$\tau_c^*$ model divided by $S_{eq}$ for the Exner-only model, to show the relative affect that that $\tau_c^*$ evolution has on equilibrium slopes. c. Equilibrium timescales for model adjustment. d. $t_{eq}$ ratio is the ratio of $t_{eq}$ for the Exner+$\tau_c^*$ model divided by $t_{eq}$ for the Exner-only model. Values are lower than 1, indicating that the $\tau_c^*$ evolution has a large influence on equilibrium timescales. e. Evolution of $\tau_c^*$.



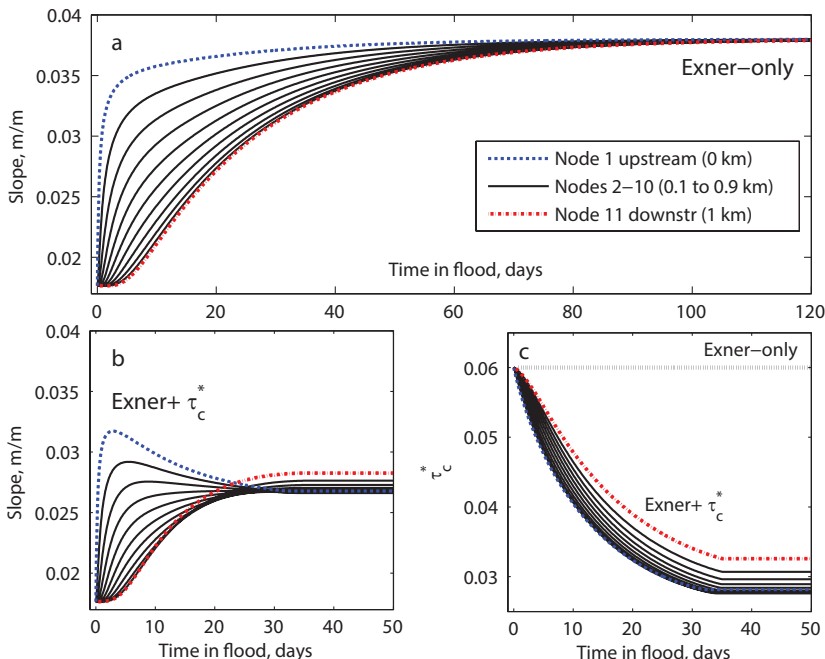

Figure 9. Spatial and temporal evolution of morphodynamic slopes, for the same models shown in Fig. 6. Slope is initially at equilibrium and responds to the 5x increase in upstream sediment supply at t=0. a. The Exner-only model initially has spatial slope variability, but evolves to a uniform new equilibrium slope. b, c. In the model with evolving $\tau_c^*$, slope and $\tau_c^*$ variability persist even at equilibrium.





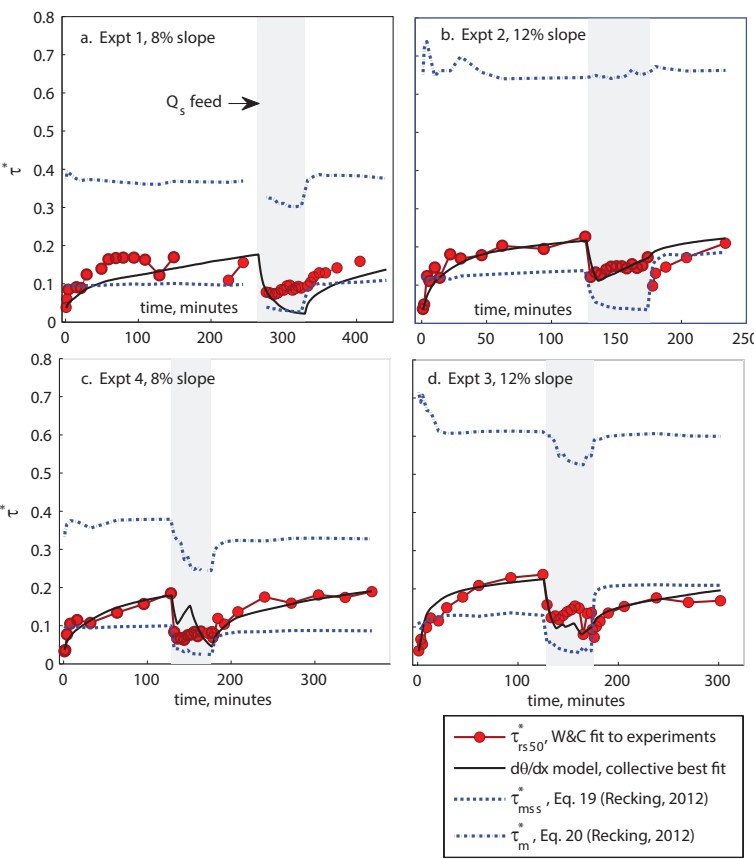

Figure 10. Comparison of experimental and best-fit model constraints on $\tau^{*}_{rs50}$, compared to proposed constraints for $D_{84}$ reference stress bounds for low and high sediment supply from Recking (2012).





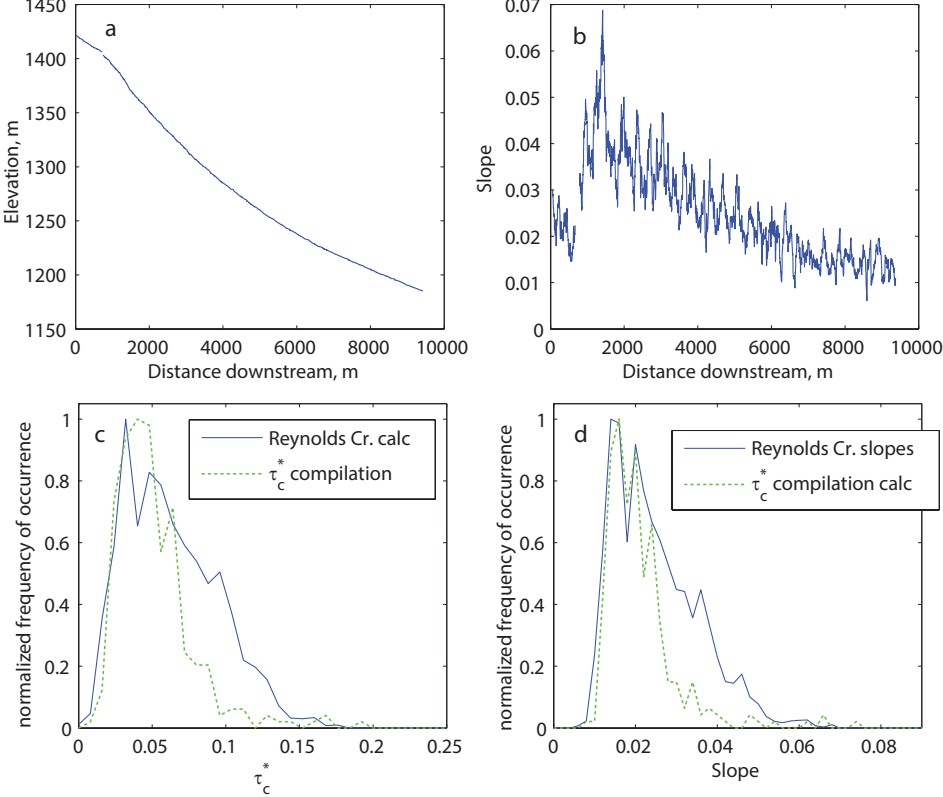

Figure 11. Comparison of Reynolds creek and $\tau_c^*$ data compilation predictions using Eq. (18) and (21). a. Longitudinal profile based on airborne Lidar (Olinde, 2015). Upstream end (distance=0) is an arbitrary location along the channel (a bridge). Data gap at 730 m is a gauging station weir; the slope steepens downstream of the weir where the valley becomes constricted by bedrock, although the bed remains almost entirely alluvial. b. Slopes averaged over 100 m reaches, illustrating reach-scale slope variability. c. Histogram of $\tau_c^*$ compilation (data shown in Fig. 1), compared to $\tau_c^*$ calculated using Eq. (21), based on Reynolds Creek 100 m slopes (panel b), and assuming qw=1 m2/s, $q_s^*$=2e-3, and f=0.1. d. Similar calculation as panel c, but using Eq. 18 to solve for slope as a function of $\tau_c^*$ from the Fig. 1 data compilation, and compared to panel b slopes.