# Peer review of "transport disequilibrium?"

_Earth Surface Dynamics, 2015_

## Referee Comment (RC1) · JM Turowski (Referee) · 8 Feb 2016

In this manuscript, the author discusses the implications of the idea that the threshold of motion is an evolving function of sediment supply. This leads to a re-definition of the threshold as a state variable in analogy to thermos dynamics. The concept is interesting and provides a fascinating change of view. My major concern is that the author does not make the above-stated re-definition explicit and uses the term threshold of motion somewhat interchangeable between the new and the old version. That makes a sometimes confusing read and can be rectified by clarifying the writing and making explicit statements. Further, I think the model is insufficiently put into physical context, and the various mechanisms that can relate sediment supply to the threshold of motion are scattered amongst the different parts of the manuscript. This can be streamlined and clarified. Some further comments to this effect follow in the next few paragraphs.
[Figure]

The physical explanations that have been proposed for the observed dependence on the threshold mostly relate to properties that the author summarized as bed state controls. Recking argued that the observed variability could at least partly be connected to changes in interlocking and armoring (see e.g., his figure 6), and Bunte et al. related the variability to bed stability, which is also dependent on properties such as interlocking. There are two possible explanations that are directly dependent on transport conditions: collective entrainment, in which moving particles mobilize stationary ones by knocking them out of their position. This mechanism has been advocated recently by Ancey and co-workers in a series of paper and demonstrated in 2D-experiments (e.g., Ancey et al. 2008; there are newer articles also available), but is highly debated by researchers working on 3D systems. The second one is the effect of fine material (sand) on the mobilization of gravel (e.g., Curran and Wilcock 2005). Although the latter could be argued to be a bed state control (the sand falls into pockets between gravel grains and therefore reduces roughness). I think the physical mechanisms that lead to the equations derived in the paper need to be better worked out and discussed, and the difference between bed state controls and direct controls of sediment supply need to be clarified. I am also not sure whether the equations actually differentiate between these two mechanisms.

The mechanism described by the author (during erosion, grains in pockets that are least stable move first, while during deposition grains stop in pockets that are most stable) could arguably be also classified as a bed state control, as it is depends on the availability of pockets of a certain degree of stability.

Further, the described mechanism in my mind only holds if either the supplied grain size distributions systematically change, or if deposition / erosion lead to systematic compaction or loosening of the bed. Consider a bed of a single grain size. By depositing a single grain, clearly it fills a pocket, but it also creates new pockets. It can be plausibly argued that the average state of the bed (roughness etc) does not change systematically in this way.

Finally, if the mechanism holds as described, there would be a feedback to roughness: deposition in stable pockets reduces the number of stable pockets, which means a smoother bed and higher flow velocity, which in turn makes each of the pockets less stable (similar to the effect of adding sand to a gravel bed, see Curran and Wilcock 2005). This would be a feedback limiting the variability of the threshold.

31 Please give some references for the statement here.

48/50 Two consecutive sentences that are both starting with 'in practice'.

55 maybe add 'typically' here

57 yes, but slope is a proxy for other parameters such as roughness, rather than a direct control

53-74 Turowski et al. 2011 demonstrate both the large temporal variability of the threshold and its control by grain and bed properties for several mountain streams. Chen and Stone 2008 explained some of the variability of measured bedload transport rates with local sub-sampling of the overall grain size distribution, leading to spatially varying thresholds of motion. This is also related to recent work on patch dynamics.

77 I am not sure whether I totally agree. See major comment.

93 comma missing after (vertical position)

136 Individual grains each have a different threshold. . .

142-143 inconsistent: does tau*_c follow a probability distribution (implying it is a random number) or is it constant?

145 and following: overuse of future tense: Progressive erosion entrains. . . grains tend to preferentially deposit. . .

147-148 This makes intuitive sense. Are there any data on this?

148-149 I am not entirely convinced by these arguments. It assumes that deposition

systematically changes bed-averaged roughness. See major comment.

158-160 Unclear why it was necessary to make this point. Please elaborate.

207 unit missing after 4.

208 does the use of 'initial' imply here that slope was changed during the experiments?

227 What does 'very low' mean here?

260 The hiding function exponent. . .

294 Which experiments? New paragraph, reference is unclear.

300-312 Curran and Wilcock 2005 should be cited somewhere here.

304 change 'with no' to 'without'.

332 Undefined abbreviation RMSD.

352 Please give the full reference.

462 Turowski et al. 2011 should be discussed in this chapter.

489-492 So, how does the model relate to the data, then?

584 There needs to be at least a brief description of Phillips' concept; it cannot be assumed that the reader is familiar with that paper.

589-605 The comparison with thermodynamics is interesting, but I wonder in how far it is novel. In the end, in river morphodynamic modelling, channels have been treated using concepts similar to state variables and state functions, they just have not been explicitly called such. Note that recently Furbish and co-workers applied concepts from statistical mechanics to bedload transport (e.g., Furbish et al. 2012, series of 4 papers in WRR and JGR).

610 This statement involves a redefinition of tau*_c, and this should be made crystal clear.
[Figure]

Fig. 4, caption: typo in matching, 3rd line.

References

Ancey, C.; Davison, A. C.; Böhm, T.; Jodeau, M. & Frey, P. Entrainment and motion of coarse particles in a shallow water stream down a steep slope, J. Fluid Mech., 2008, 595, 83-114

Chen, L. & Stone, M. C. Influence of bed material size heterogeneity on bedload transport uncertainty, Water Resources Research, 2008, 44, W01405

Curran, J. C. & Wilcock, P. R. Effect of sand supply on transport rates in a gravel-bed channel J. Hydr. Eng., 2005, 131, 961-967

Furbish, D. J.; Haff, P. K.; Roseberry, J. C. & Schmeeckle, M. W. A probabilistic description of the bed load sediment flux: 1. Theory, Journal of Geophysical Research, 2012, 117, F03031

Turowski, J.M., A. Badoux, D. Rickenmann, 2011, Start and end of bedload transport in gravel bed rivers, Geophysical Research Letters, 38, L04401, doi: 10.1029/2010GL046558

---

## Referee Comment (RC2) · Anonymous Referee #2 · 12 Apr 2016

Review of 'Gravel threshold of motion: A state function of sediment transport disequilibrium' Earth Surface Dynamics (esurf-2015-52) Joel P. L. Johnson

This paper uses flume experiments and a morphodynamic model to assess the impact that sediment supply has on the evolution of thresholds of motion. The topic of the paper is of interest to readers with some interesting findings that are applicable to the wider discipline. However at present the paper is quite, long, 'dense' and difficult to read in parts meaning that the novelty of the paper is somewhat lost in places.

The main comment I feel which needs addressing in this paper is the lack of emphasis on the physical underpinnings of how sediment supply affects the thresholds of motion. Whilst the author makes reference to the bed state conditions in the introduction he does not really follow those through in terms of the implications of his findings. This

currently leaves the reader wanting more detail in this regard. There are many papers which talk about the effects of both bed state in terms of structure as well as sand content on entrainment thresholds. I think the latter is particularly important for this paper and the author could look at the following papers as a starting point.

Curran, J.C. and Wilcock, P.R. (2005). Effect of sand supply on transport rates in gravel bed channels. Journal of Hydraulic Engineering. 131:961-967 Ikeda, H. and Iseya, F. (1988). Experimental study of heterogeneous sediment transport. Environmental Research Centre Paper 12. University of Tsukuba; Japan. Jackson, W. L., and Beschta, R.L. (1984). Influences of increased sand delivery on the morphology of sand and gravel channels. Journal of the American Water Resources Association. 20; 527–533.

I also feel the paper could benefit from being shortened as it is currently quite long and loses focus in places. Detailed comments are also given below.

Line 83- I am not sure I agree with the statement that is still only believed to be controlled by grain parameters. There is an increasing recognition that, as the author alludes, bed state controls are also important. I think at the very least this should be recognised in the current text and references made to the large body of work relating to the impact of structure on bed stability. How does this also link to the concept of mobile armours? You go on to mention this in lines 153-157 so this section could be reorganised?

Line 93- comma missing after vertical position

Lines 123- 131- this section is clumsy and needs re-writing

Line 141- 143- does this not assume that the bed state does not change? You could have the same overall flux of sediment but the surface structure may change and hence the distribution of threshold stress will thus change as the bed is more stable?

Line 155- consider revision of little additional decrease

Lines 158 – 167 – I think if you are using the terms interchangeably throughout the

paper then there is no need for this paragraph at all.

Line 185- should be dimensional not dimensionally

Line 189 – move 'Ar is an optional dimensionless armouring parameter, described further below' to line 198 where you talk about Ar

Line 189- I think the Ar should be defined as it can have different definitions

Line 202- this sentence does not make sense- do you mean large grains rather than large range?

Line 205- although this paper is concentrating on step pool sequences perhaps something to consider later on in the paper is how applicable these results are to gravel bed rivers more broadly e.g. at lower slopes?

Line 207 – unit missing after flume length

Line 226- can you be more specific- how much erosion?

Line 227 – what does 'very low' mean? Can you quantify?

Line 228 – why was this feed rate chosen? What was this rate in comparison to the initial transport rates?

Lines 243- consider deleting to GSDs compared

Lines 237- why was the Wilcock and Crowe model specifically used?

Lines 237- 265- can this section be shortened? Why not just reference the W&CM highlighting the changes you made to it?(lines 262-263)

Lines 313 -316 – what was your GSD? This is important if you are beginning to duscss sand content and the mechanisms by which sediment feed rate affects initiation of motion? Also in line 313 you mention that the % of grains smaller than 2mm was very small bu tin lines 316 you say 2.8mm was your smallest grain fraction?

Lines 332 – define RMSD

Lines 336 – 344 – this is an interesting finding but what are the implications of this in terms of bed state?

Line 352 – need full reference to Parker

Line 473- change stresses to stress

Line 474- I am not sure they are comparable are they? Again thinking in terms of the relative effects of bed structure and implications of grain size, structure and thresholds of motion would D50 and D84 be expected to behave the same?

Line 475- what do you mean by 'fairly comparable'?

Lines 503-506- I think this is one of the places where a better physical explanation behind the findings would be useful

Lines 530 – this section is supposed to be linked to system memory but I find it hard to distinguish this and a much more explicit link needs to be made.

Lines 545-546 – I would re-write to avoid asking a rhetorical question

Lines 576 – whilst I find this section an interesting concept I think it could be shortened a lot given the paper is already quite long.

Line 584- expand upon the work of Phillips (2007)

Line 625 - I would re-write to avoid asking a rhetorical question

---

## Editor Comment (EC1) · D.R Parsons (Editor) · 22 Jun 2016

Dear Joel, The two reviewer reports are available for your manuscript. Both raise a few areas where they feel that the manuscript could be improved. I will be happy to recommend that the manuscript be accepted subject to the minor revisions based on these comments. If you could respond to the reviews and supply a rebuttal that details the changes made to the script that would be appreciated. Regards Dan

---

## Referee Comment (RC3) · Anonymous Referee #3 · 29 Jun 2016

I believe this is an overall excellent piece of work, written by an expert in the field. The issue of sediment transport is a long studied problem and much attention has been paid to traditional criteria, such as Shield's critical shear stresses (as the author notes himself). There are a number of problems using such criteria - as the author mentions in his work (also demonstrated in Fig. 1). However, the author still chooses to deploy this criterion focusing on the fact that data scatter (e.g. in Fig.1) is due to a range of factors, however omitting to discuss its inability to represent the rich dynamics of grain transport, as recent research has shown (Schemeeckle et al. 2003, Diplas et al. 2008).

The major novelty of the present work lays in the presentation of a state function for the description of sediment transport, which is a very much welcomed contribution as a conceptual approach. However, there is a significant concern (to this reviewer) over the suitability of the Shield's shear stress as parameter to be used in this model. Would not

other more criteria that capture the full range of grain dynamics, such as instantaneous hydrodynamic forces near the bed or even better the impulse/energy content of flow structures, be more suitable as model parameters? Of course such analysis may offer enough new material for another (and perhaps more impactful) publication, but yet it may be useful to add a note about this on the discussion section.

Another, minor issue is with the interpretation of the data analysis. In particular, is there no better measure to assess the "amount of information embedded" between two variables than R2? R2 is rather demonstrative of the strength of association between two variables.

---

## Author Comment (AC1) · 31 Jul 2016

In this manuscript, the author discusses the implications of the idea that the threshold of motion is an evolving function of sediment supply. This leads to a re-definition of the threshold as a state variable in analogy to thermos dynamics. The concept is interesting and provides a fascinating change of view. My major concern is that the author does not make the above-stated re-definition explicit and uses the term threshold of motion somewhat interchangeable between the new and the old version. That makes a sometimes confusing read and can be rectified by clarifying the writing and making explicit statements. Further, I think the model is insufficiently put into physical context, and the various mechanisms that can relate sediment supply to the threshold of motion are scattered amongst the different parts of the manuscript. This can be streamlined and clarified. Some further comments to this effect follow in the next few paragraphs.

Thank you for the constructive comments. Following the recommendation that the physical processes causing changes to thresholds of motion be described in more detail and combined in one place, the biggest change I made to the manuscript is moving parts of two sections that discuss previous work on evolving thresholds of motion—part of previous section 3.2 that discussed the sand dependence of reference stresses in the Wilcock and Crowe (2003) model, and also most of previous section 4.1 ("Comparison to previous work"), which discussed Recking (2012) relations—into the introduction. These are now section 1.1. In this way I have one section that better describes the many various physical controls on thresholds of motion.

I have also made the "redefinition" of thresholds more explicit, in two ways. First, I have slightly modified my notation: in the previous version I only used $\tau_c^*$ as the threshold variable. In the new version, I have added variable $\tau_{c(q_s)}^*$ to specifically indicate the new sediment flux-dependent model. In addition, I specifically describe the model as a redefinition of the concept of thresholds of motion (new lines 641, 646, 735).

Because of how Word changes line numbers in the "track changes" version of the manuscript, I note that the line numbers refer to the revised manuscript that does not show all of the edits.

The physical explanations that have been proposed for the observed dependence on the threshold mostly relate to properties that the author summarized as bed state controls. Recking

argued that the observed variability could at least partly be connected to changes in interlocking and armoring (see e.g., his figure 6), and Bunte et al. related the variability to bed stability, which is also dependent on properties such as interlocking. There are two possible explanations that are directly dependent on transport conditions: collective entrainment, in which moving particles mobilize stationary ones by knocking them out of their position. This mechanism has been advocated recently by Ancey and co-workers in a series of paper and demonstrated in 2D-experiments (e.g., Ancey et al. 2008; there are newer articles also available), but is highly debated by researchers working on 3D systems. The second one is the effect of fine material (sand) on the mobilization of gravel (e.g., Curran and Wilcock 2005). Although the latter could be argued to be a bed state control (the sand falls into pockets between gravel grains and therefore reduces roughness). I think the physical mechanisms that lead to the equations derived in the paper need to be better worked out and discussed, and the difference between bed state controls and direct controls of sediment supply need to be clarified. I am also not sure whether the equations actually differentiate between these two mechanisms.

I have worked explicit descriptions of these processes and citations into the manuscript, both in section 1.1 where previous work is reviewed, and also section 2.1 where the new model is presented conceptually.

The mechanism described by the author (during erosion, grains in pockets that are least stable move first, while during deposition grains stop in pockets that are most stable) could arguably be also classified as a bed state control, as it is depends on the availability of pockets of a certain degree of stability.

Good point. I am now more clear that my categorizations of threshold controls are not absolute, that the controls are interrelated, and that many controls could be categorized in different ways (section 1.1; new lines 84-87, 92-95 for example).

Further, the described mechanism in my mind only holds if either the supplied grain size distributions systematically change, or if deposition / erosion lead to systematic compaction or loosening of the bed. Consider a bed of a single grain size. By depositing a single grain, clearly it fills a pocket, but it also creates new pockets. It can be plausibly argued that the average state of the bed (roughness etc) does not change systematically in this way.

Finally, if the mechanism holds as described, there would be a feedback to roughness: deposition in stable pockets reduces the number of stable pockets, which means a smoother bed and higher flow velocity, which in turn makes each of the pockets less stable (similar to the effect of adding sand to a gravel bed, see Curran and Wilcock 2005). This would be a feedback limiting the variability of the threshold.

Good points. To address this, I have expanded the description of feedbacks in section 2.1 (the conceptual model). I now explicitly say in this section that there are physical limits of how much bed roughness and other controls can change (new lines 260-265). These limits were already built into the model equations before, but were previously not described well enough conceptually.

31 Please give some references for the statement here.

I added five references (new line 29-31)

48/50 Two consecutive sentences that are both starting with 'in practice'.

Rearranged and combined sentences to remove the repetition.

55 maybe add 'typically' here

Done (new line 60).

57 yes, but slope is a proxy for other parameters such as roughness, rather than a direct control

I agree; this is now stated directly (new lines 63-66).

53-74 Turowski et al. 2011 demonstrate both the large temporal variability of the threshold and its control by grain and bed properties for several mountain streams. Chen and Stone 2008 explained some of the variability of measured bedload transport rates with local sub-sampling of the overall grain size distribution, leading to spatially varying thresholds of motion. This is also related to recent work on patch dynamics.

I have added description and reference to these works, and also now state that patches influencing thresholds of motion and transport (new lines 66-69, 99-101).

77 I am not sure whether I totally agree. See major comment.

I have now clarified how I categorize controls on thresholds of motion, simply for the sake of describing controls in an organized manner. I have also added a separate category of sediment flux controls (new lines 84-87, 142-170).

93 comma missing after (vertical position)
Added comma

136 Individual grains each have a different threshold…
Done (cut the word "will")

142-143 inconsistent: does tau*_c follow a probability distribution (implying it is a random number) or is it constant?

I have clarified the relationship between distributions of threshold values for a population of grains on the bed surface, and the single threshold value that would best describe transport when applied in a bedload transport equation (new lines 229-239).

145 and following: overuse of future tense: Progressive erosion entrains… grains tend to preferentially deposit…

I have changed writing to be present tense, here and elsewhere.

147-148 This makes intuitive sense. Are there any data on this?

I wish there were, but I am unaware of data showing this. I hope to collect flume data on this in the future. I have addressed this comment by adding "I assume" to make it clear that this is an assumption of the model (new lines 244-246).

148-149 I am not entirely convinced by these arguments. It assumes that deposition systematically changes bed-averaged roughness. See major comment.

The reviewer is right, it does generally assume that bed-averaged roughness changes. I now clarify in this section that there are limits to how far thresholds of motion can evolve (new lines 260-265).

158-160 Unclear why it was necessary to make this point. Please elaborate.
I have cut this part.

207 unit missing after 4.
Added.

208 does the use of 'initial' imply here that slope was changed during the experiments?

I have rearranged text to now have the callout to Figure 2 sooner, which shows (minimal) slope evolution during the experiments. The data were included in the Figure in the previous version, but it was less clear. (new lines 319-325).

227 What does 'very low' mean here?
I have reworded the text to say that flux dropped by approximately 3 orders of magnitude, and also have a callout here to Figure 2a that shows how the transport rate changed through time. (new lines 343-344).

260 The hiding function exponent…
Added "The" as requested.

294 Which experiments? New paragraph, reference is unclear.
Edited to be clear what data is being talked about (new lines 405-407).

300-312 Curran and Wilcock 2005 should be cited somewhere here.
Added (new line 414).

304 change 'with no' to 'without'.
This wording was cut during editing.

332 Undefined abbreviation RMSD.
Done (new line 433)

352 Please give the full reference.
Done (new line 451)

462 Turowski et al. 2011 should be discussed in this chapter.

I now reference this work in multiple places in the manuscript. This particular section of text has been moved to the introduction.

489-492 So, how does the model relate to the data, then?
I have expanded this paragraph a bit to more to explain how the model can be consistent with and explains previous observations of Recking (2012) and Bunte et al. (2013), but at the same time the model does not depend directly on sediment flux, but on changes in sediment flux. (new lines 561-570).

584 There needs to be at least a brief description of Phillips' concept; it cannot be assumed that the reader is familiar with that paper.
Done, new lines 648-652.

589-605 The comparison with thermodynamics is interesting, but I wonder in how far it is novel. In the end, in river morphodynamic modelling, channels have been treated using concepts similar to state variables and state functions, they just have not been explicitly called such. Note that recently Furbish and co-workers applied concepts from statistical mechanics to bedload transport (e.g., Furbish et al. 2012, series of 4 papers in WRR and JGR).
While I thought the previous version acknowledged that similar ideas have been implicitly used, I now state this directly (new lines 647-648). I now cite the Furbish work earlier in the manuscript (new lines 217-221); it does not bear direct relevance to rate and state variables, though is excellent work applying ideas from physics.

610 This statement involves a redefinition of tau*_c, and this should be made crystal clear.
I now explicitly state that the threshold of motion is defined as a state variable (new line 641, 646, 677).

Fig. 4, caption: typo in matching, 3rd line.
Corrected.

References
Ancey, C.; Davison, A. C.; Böhm, T.; Jodeau, M. & Frey, P. Entrainment and motion of coarse particles in a shallow water stream down a steep slope, J. Fluid Mech., 2008, 595, 83-114

Chen, L. & Stone, M. C. Influence of bed material size heterogeneity on bedload transport uncertainty, Water Resources Research, 2008, 44, W01405

Curran, J. C. & Wilcock, P. R. Effect of sand supply on transport rates in a gravel-bed channel J. Hydr. Eng., 2005, 131, 961-967

Furbish, D. J.; Haff, P. K.; Roseberry, J. C. & Schmeeckle, M. W. A probabilistic description of the bed load sediment flux: 1. Theory, Journal of Geophysical Research, 2012, 117, F03031

Turowski, J.M., A. Badoux, D. Rickenmann, 2011, Start and end of bedload transport in gravel bed rivers, Geophysical Research Letters, 38, L04401, doi: 10.1029/2010GL046558

---

## Author Comment (AC2) · 31 Jul 2016

Review of 'Gravel threshold of motion: A state function of sediment transport disequilibrium'
Earth Surface Dynamics (esurf-2015-52) Joel P. L. Johnson

This paper uses flume experiments and a morphodynamic model to assess the impact that sediment supply has on the evolution of thresholds of motion. The topic of the paper is of interest to readers with some interesting findings that are applicable to the wider discipline. However at present the paper is quite, long, 'dense' and difficult to read in parts meaning that the novelty of the paper is somewhat lost in places. The main comment I feel which needs addressing in this paper is the lack of emphasis on the physical underpinnings of how sediment supply affects the thresholds of motion. Whilst the author makes reference to the bed state conditions in the introduction he does not really follow those through in terms of the implications of his findings. This currently leaves the reader wanting more detail in this regard. There are many papers which talk about the effects of both bed state in terms of structure as well as sand content on entrainment thresholds. I think the latter is particularly important for this paper and the author could look at the following papers as a starting point.

I appreciate the comments, and have worked to address them by reorganizing and expanding on the physical explanations for why thresholds of motion can change over time. Following these comments, I have streamlined some sections and shortened the manuscript wherever I felt possible. However, because of additional clarifications and explanations requested by the reviewers the manuscript is slightly (~4.5%) longer as measured by text, but has 1 fewer figures. More importantly, I believe it is more clear and focused.

Following the recommendations of both Turowski and Reviewer2 that the physical processes causing changes to thresholds of motion be described in more detail and combined in one place, the biggest change I made to the manuscript is moving parts of two sections that discuss previous work on evolving thresholds of motion—part of previous section 3.2 that discussed the sand dependence of reference stresses in the Wilcock and Crowe (2003) model, and also most of previous section 4.1 ("Comparison to previous work"), which discussed Recking (2012) relations—into the introduction. These are now section 1.1. In this way I have one section that better describes the many various physical controls on thresholds of motion. This topic is presented in section 1.1, and also 2.1. While the effect of sand on thresholds of motion was addressed in the previous version, it is now discussed much more prominently near the beginning of the manuscript.

Because of concerns over length brought up by Reviewer2, I cut one figure (previous Figure 11) and the section of text that went along with it. I removed this part because, while interesting, I felt like it was less central to the science than the other parts of the work. The other 10 figures are essentially unchanged.

Curran, J.C. and Wilcock, P.R. (2005). Effect of sand supply on transport rates in gravel bed channels. Journal of Hydraulic Engineering. 131:961-967

Ikeda, H. and Iseya, F.(1988). Experimental study of heterogeneous sediment transport. Environmental Research Centre Paper 12. University of Tsukuba; Japan.

Jackson, W. L., and Beschta,R.L. (1984). Influences of increased sand delivery on the morphology of sand and gravel channels. Journal of the American Water Resources Association. 20; 527–533.

I now reference these papers in regards to sand supply and thresholds of motion.

I also feel the paper could benefit from being shortened as it is currently quite long and loses focus in places. Detailed comments are also given below.

I have tried hard to improve the paper by following the comments of all of the reviewers. This includes adding material to explain many points further, hopefully making the manuscript less dense. While I have also cut text (in particular Figure 11 and the text along with it), unfortunately the manuscript is now slightly longer as measured by text. However, I also believe that it is more clear and understandable. The revised manuscript better guides the reader and explains why certain things are being presented, hopefully helping it keep its focus (for example, new lines 79-82, 203-211, 386-389, and 545-548).

Because of how Word changes line numbers in the "track changes" version of the manuscript, I note that the line numbers refer to the revised manuscript that does not show all of the edits.

Line 83- I am not sure I agree with the statement that is still only believed to be controlled by grain parameters. There is an increasing recognition that, as the author alludes, bed state controls are also important. I think at the very least this should be recognised in the current text and references made to the large body of work relating to the impact of structure on bed stability. How does this also link to the concept of mobile armours? You go on to mention this in lines 153-157 so this section could be reorganised?

Section 1.1 now more clearly lays out the relations and overlap between what I am categorizing as grain controls and bed state controls (new lines 84-87). I now specifically address and reference armoring in relation to both of these categories (new lines 92-95). Because space is limited I do not expand on the differences between mobile and static armors.

Line 93- comma missing after vertical position
Added comma.

Lines 123- 131- this section is clumsy and needs re-writing
I edited this section to use active voice and to be less awkward. (new lines 203-211).

Line 141- 143- does this not assume that the bed state does not change? You could have the same overall flux of sediment but the surface structure may change and hence the distribution of threshold stress will thus change as the bed is more stable?
I rearranged this section to have the mention of steady state at the end of the section rather than at the beginning, to more clearly explain how the proposed feedbacks work. I also expanded specifically on the case of steady state threshold of motion based on this comment: Yes, the threshold could evolve under constant flux, but because sediment transport rate and the threshold of motion are directly linked, a change in threshold would change the transport rate (new lines 268-274).

Line 155- consider revision of little additional decrease
Changed wording.

Lines 158 – 167 – I think if you are using the terms interchangeably throughout the paper then there is no need for this paragraph at all.

I got rid of some of this paragraph, and shorted and moved part of it to elsewhere (new lines 52-57). I believe it is important to explicitly state what threshold stresses and reference stresses are, and to justify using them interchangeably.

Line 185- should be dimensional not dimensionally
Changed.

Line 189 – move 'Ar is an optional dimensionless armouring parameter, described further below' to line 198 where you talk about Ar. I think the Ar should be defined as it can have different definitions.

Done; I moved text around and also expanded on this portion, to explain and justify this model parameter (new lines 307-316).

Line 202- this sentence does not make sense- do you mean large grains rather than large range?
That is exactly what I meant; I somehow put the wrong word. I'm glad the reviewer was able to figure out the intention of the sentence.

Line 205- although this paper is concentrating on step pool sequences perhaps something to consider later on in the paper is how applicable these results are to gravel bed rivers more broadly e.g. at lower slopes?
I have added a statement to the discussion that the model parameters were calibrated to these particular steep slopes, and that future work is required to validate the model over a broader range of parameter space. (new lines 635-639).

Line 207 – unit missing after flume length
Done, added m.

Line 226- can you be more specific- how much erosion?
In the interest of clarity and length, I edited the text to focus on the bed responses (coarsening and roughening and sediment transport rates) that matter more for my analysis; in doing so I cut the explicit mention of erosion. (new lines 343-344). Erosion amounts are less informative and somewhat different for the different experiments.

Line 227 – what does 'very low' mean? Can you quantify?
I have reworded the text to say that flux dropped by approximately 3 orders of magnitude, and also have a callout to Figure 2a that shows the transport rate changes through time. (new lines 341-344).

Line 228 – why was this feed rate chosen? What was this rate in comparison to the initial transport rates?
I added an explanation of why the feed rate was chosen—"this feed rate was chosen to be similar to the high initial transport rates (Fig. 2a), while not so high as to inhibit morphodynamic feedbacks by fully burying the stabilized bed surfaces." (new lines 347-349).

Lines 243- consider deleting to GSDs compared
Done

Lines 237- why was the Wilcock and Crowe model specifically used?
I now give the specific reasons that I used this model: because it can account for both surface grain size changes and also let me evaluate whether sand supply can explain the experimental transport trends. (new lines 359-365).

Lines 237- 265- can this section be shortened? Why not just reference the W&CM highlighting the changes you made to it?(lines 262-263)

I considered cutting some of the equations that are Wilcock and Crowe (2003) model, but in the end decided to leave them in. I believe that cutting Eq. 13 and 14 (previously 10 and 11), which show what nondimensional bedload transport rate means and how thresholds of motion actually go into the transport relation, would make the paper more difficult to understand for most readers, especially those not intimately familiar with W&CM. Also, since I made changes to equations 15 and 16 (previously 12 and 13) I would have to leave those equations in the manuscript; the length of writing actually cut would be pretty small.

Lines 313 -316 – what was your GSD? This is important if you are beginning to duscss sand content and the mechanisms by which sediment feed rate affects initiation of motion? Also in line 313 you mention that the % of grains smaller than 2mm was very small bu tin lines 316 you say 2.8mm was your smallest grain fraction?

I now clarify and describe more completely the full grain size distributions used in the flume experiments; relevant here is that the smallest size class used in the experiments had a D16 of 2.0 mm, D50=2.4 mm, and D84 of 2.8 mm (new lines 327-330). I clarify that this was the size class used for the calculations of sand fraction (new lines 414-422).

Lines 332 – define RMSD
Done (new line 433)

Lines 336 – 344 – this is an interesting finding but what are the implications of this in terms of bed state?
I now give a suggestion of why my model was seemingly insensitive to having combinations of D84, D50, D16 and bed roughness included as another parameter in the model—it may suggest that net erosion and deposition were more important over the range of parameter space explored in the experiments (new lines 440-442).

Line 352 – need full reference to Parker
Done, new line 451.

Line 473- change stresses to stress
Done

Line 474- I am not sure they are comparable are they? Again thinking in terms of the relative effects of bed structure and implications of grain size, structure and thresholds of motion would D50 and D84 be expected to behave the same?
I clarified the writing; I was not trying to suggest that D50 and D84 thresholds would necessarily be or should be expected to be interchangeable. I simply make the point that Recking's relation is not too far off from my thresholds of motion, although the R^2 value is still low (new lines 552-560).

Line 475- what do you mean by 'fairly comparable'?
Reworded to say I do not expect these different threshold measured to be equivalent (new lines 553-554).

Lines 503-506- I think this is one of the places where a better physical explanation behind the findings would be useful
Good point. I added a substantial amount of text. First, I now acknowledge that future work would be required to really determine specific process linkages explaining asymmetry in

aggradation vs degradation effects on thresholds of motion (new lines 587-589). Second, I present a hypothesis that could be tested with future work about the differences in deposition, erosion and roughness evolution (new lines 589-606).

Lines 530 – this section is supposed to be linked to system memory but I find it hard to distinguish this and a much more explicit link needs to be made.

While the section talked about system memory, it also covered other topics. In the interest of article length I removed the section (old section 4.3 in the previous manuscript, and also figure 11) and also cut most of the content. I did however keep and expand slightly on parts related to memory, these are now at new lines 628-639.

Lines 545-546 – I would re-write to avoid asking a rhetorical question

This sentence was cut during editing.

Lines 576 – whilst I find this section an interesting concept I think it could be shortened a lot given the paper is already quite long.

I shortened this section by roughly 23% through editing (new section 4.2). However, I feel like it is an important idea to explain thoroughly, and I really do not want to remove more of it. I did cut substantial portions of old section 4.3 in the previous manuscript, and also old figure 11, in order to keep the manuscript focused and not even longer.

Line 584- expand upon the work of Phillips (2007)

Done, new lines 649-652.

Line 625 - I would re-write to avoid asking a rhetorical question

Done. I also edited out some other rhetorical questions in the manuscript.

---

## Author Comment (AC3) · 31 Jul 2016

Dear Joel, The two reviewer reports are available for your manuscript. Both raise a few areas where they feel that the manuscript could be improved. I will be happy to recommend that the manuscript be accepted subject to the minor revisions based on these comments. If you could respond to the reviews and supply a rebuttal that details the changes made to the script that would be appreciated. Regards Dan

Thank you for the opportunity to revise the manuscript. Following the comments and recommendations of all three reviewers, I have rearranged parts of the manuscript, added more detail to make parts of it be more organized and less abstract, and have also added recommended citations. Following reviewer recommendations, I have streamlined some sections and shortened the manuscript wherever I felt possible. Because of additional clarifications and explanations requested by the reviewers the manuscript is slightly (~4.5%) longer as measured by text, but has 1 fewer figures. More importantly, I believe it is more clear and focused.

Following the recommendations of both Turowski and Reviewer2 that the physical processes causing changes to thresholds of motion be described in more detail and combined in one place, the biggest change I made to the manuscript is moving parts of two sections that discuss previous work on evolving thresholds of motion—part of previous section 3.2 that discussed the sand dependence of reference stresses in the Wilcock and Crowe (2003) model, and also most of previous section 4.1 ("Comparison to previous work"), which discussed Recking (2012) relations—into the introduction. These are now section 1.1. In this way I have one section that better describes the many various physical controls on thresholds of motion.

Because of concerns over length brought up by Reviewer2, I cut one figure (previous Figure 11) and the section of text that went along with it. I removed this part because, while interesting, I felt like it was less central to the science than the other parts of the work. The other 10 figures are essentially unchanged.

**JM Turowski (Referee)**
turowski@gfz-potsdam.de

In this manuscript, the author discusses the implications of the idea that the threshold of motion is an evolving function of sediment supply. This leads to a re-definition of the threshold as a state variable in analogy to thermos dynamics. The concept is interesting and provides a fascinating change of view. My major concern is that the author does not make the above-stated re-definition explicit and uses the term threshold of motion somewhat interchangeable between

the new and the old version. That makes a sometimes confusing read and can be rectified by clarifying the writing and making explicit statements. Further, I think the model is insufficiently put into physical context, and the various mechanisms that can relate sediment supply to the threshold of motion are scattered amongst the different parts of the manuscript. This can be streamlined and clarified. Some further comments to this effect follow in the next few paragraphs.

As recommended, I have moved some of the "scattered" explanations to section 1.1, and expanded the physical explanations. I have also made the "redefinition" of thresholds more explicit, in two ways. First, I have slightly modified my notation: in the previous version I only used $\tau_c^*$ as the threshold variable. In the new version, I have added variable $\tau_{c(q_s)}^*$ to specifically indicate the new sediment flux-dependent model. In addition, I specifically describe the model as a redefinition of the concept of thresholds of motion (new lines 641, 646, 735).

The physical explanations that have been proposed for the observed dependence on the threshold mostly relate to properties that the author summarized as bed state controls. Recking argued that the observed variability could at least partly be connected to changes in interlocking and armoring (see e.g., his figure 6), and Bunte et al. related the variability to bed stability, which is also dependent on properties such as interlocking. There are two possible explanations that are directly dependent on transport conditions: collective entrainment, in which moving particles mobilize stationary ones by knocking them out of their position. This mechanism has been advocated recently by Ancey and co-workers in a series of paper and demonstrated in 2D-experiments (e.g., Ancey et al. 2008; there are newer articles also available), but is highly debated by researchers working on 3D systems. The second one is the effect of fine material (sand) on the mobilization of gravel (e.g., Curran and Wilcock 2005). Although the latter could be argued to be a bed state control (the sand falls into pockets between gravel grains and therefore reduces roughness). I think the physical mechanisms that lead to the equations derived in the paper need to be better worked out and discussed, and the difference between bed state controls and direct controls of sediment supply need to be clarified. I am also not sure whether the equations actually differentiate between these two mechanisms.

I have worked explicit descriptions of these processes and citations into the manuscript, both in section 1.1 and also 2.1, where previous work is reviewed, and also where the new model is presented conceptually.

The mechanism described by the author (during erosion, grains in pockets that are least stable move first, while during deposition grains stop in pockets that are most stable) could arguably be also classified as a bed state control, as it is depends on the availability of pockets of a certain degree of stability.

Good point. I am now more clear that my categorizations of threshold controls are not absolute, that the controls are interrelated, and that many controls could be categorized in different ways (section 1.1; new lines 84-87, 92-95 for example).

Further, the described mechanism in my mind only holds if either the supplied grain size distributions systematically change, or if deposition / erosion lead to systematic compaction or loosening of the bed. Consider a bed of a single grain size. By depositing a single grain, clearly

it fills a pocket, but it also creates new pockets. It can be plausibly argued that the average state of the bed (roughness etc) does not change systematically in this way.

Finally, if the mechanism holds as described, there would be a feedback to roughness: deposition in stable pockets reduces the number of stable pockets, which means a smoother bed and higher flow velocity, which in turn makes each of the pockets less stable (similar to the effect of adding sand to a gravel bed, see Curran and Wilcock 2005). This would be a feedback limiting the variability of the threshold.

Good points. To address this, I have expanded the description of feedbacks in section 2.1 (the conceptual model). I now explicitly say in this section that there are physical limits of how much bed roughness and other controls can change (new lines 260-265). These limits were already built into the model equations before, but were previously not described well enough conceptually.

31 Please give some references for the statement here.

I added five references (new line 29-31)

48/50 Two consecutive sentences that are both starting with 'in practice'.

Rearranged and combined sentences to remove the repetition.

55 maybe add 'typically' here

Done (new line 60).

57 yes, but slope is a proxy for other parameters such as roughness, rather than a direct control

I agree; this is now stated directly (new lines 63-66).

53-74 Turowski et al. 2011 demonstrate both the large temporal variability of the threshold and its control by grain and bed properties for several mountain streams. Chen and Stone 2008 explained some of the variability of measured bedload transport rates with local sub-sampling of the overall grain size distribution, leading to spatially varying thresholds of motion. This is also related to recent work on patch dynamics.

I have added description and reference to these works, and also now state that patches influencing thresholds of motion and transport (new lines 66-69, 99-101).

77 I am not sure whether I totally agree. See major comment.

I have now clarified how I categorize controls on thresholds of motion, simply for the sake of describing controls in an organized manner. I have also added a separate category of sediment flux controls (new lines 84-87, 142-170).

93 comma missing after (vertical position)
Added comma

136 Individual grains each have a different threshold…
Done (cut the word "will")

142-143 inconsistent: does tau*_c follow a probability distribution (implying it is a random number) or is it constant?

I have clarified the relationship between distributions of threshold values for a population of grains on the bed surface, and the single threshold value that would best describe transport when applied in a bedload transport equation (new lines 229-239).

145 and following: overuse of future tense: Progressive erosion entrains… grains tend to preferentially deposit…

I have changed writing to be present tense, here and elsewhere.

147-148 This makes intuitive sense. Are there any data on this?

I wish there were, but I am unaware of data showing this. I have addressed this comment by adding "I assume" to make it clearer that this is an assumption of the model (new lines 244-246).

148-149 I am not entirely convinced by these arguments. It assumes that deposition systematically changes bed-averaged roughness. See major comment.

The reviewer is right, it does generally assume that bed-averaged roughness changes. I now clarify in this section that there are limits to how far thresholds of motion can evolve (new lines 260-265).

158-160 Unclear why it was necessary to make this point. Please elaborate.
I have cut this part.

207 unit missing after 4.
Added.

208 does the use of 'initial' imply here that slope was changed during the experiments?

I have rearranged text to now have the callout to Figure 2 sooner, which shows (minimal) slope evolution during the experiments. The data were included in the Figure in the previous version, but it was less clear. (new lines 319-325).

227 What does 'very low' mean here?
I have reworded the text to say that flux dropped by approximately 3 orders of magnitude, and also have a callout here to Figure 2a that shows how the transport rate changed through time. (new lines 343-344).

260 The hiding function exponent…

Added "The" as requested.

294 Which experiments? New paragraph, reference is unclear.
Edited to be clear what data is being talked about (new lines 405-407).

300-312 Curran and Wilcock 2005 should be cited somewhere here.
Added (new line 414).

304 change 'with no' to 'without'.
This wording was cut during editing.

332 Undefined abbreviation RMSD.
Done (new line 433)

352 Please give the full reference.
Done (new line 451)

462 Turowski et al. 2011 should be discussed in this chapter.
I now reference this work in multiple places in the manuscript. This particular section of text has been moved to the introduction.

489-492 So, how does the model relate to the data, then?
I have expanded this paragraph a bit to more to explain how the model can be consistent with and explains previous observations of Recking (2012) and Bunte et al. (2013), but at the same time the model does not depend directly on sediment flux, but on changes in sediment flux. (new lines 561-570).

584 There needs to be at least a brief description of Phillips' concept; it cannot be assumed that the reader is familiar with that paper.
Done, new lines 648-652.

589-605 The comparison with thermodynamics is interesting, but I wonder in how far it is novel. In the end, in river morphodynamic modelling, channels have been treated using concepts similar to state variables and state functions, they just have not been explicitly called such. Note that recently Furbish and co-workers applied concepts from statistical mechanics to bedload transport (e.g., Furbish et al. 2012, series of 4 papers in WRR and JGR).
While I thought the previous version acknowledged that similar ideas have been implicitly used, I now state this directly (new lines 647-648). I now cite the Furbish work earlier in the manuscript (new lines 217-221); it does not bear direct relevance to rate and state variables, though is excellent work applying ideas from physics.

610 This statement involves a redefinition of tau*_c, and this should be made crystal clear.
I now explicitly state that the threshold of motion is defined as a state variable (new line 641, 646, 677).

Fig. 4, caption: typo in matching, 3rd line.
Corrected.

References

Ancey, C.; Davison, A. C.; Böhm, T.; Jodeau, M. & Frey, P. Entrainment and motion of coarse particles in a shallow water stream down a steep slope, J. Fluid Mech., 2008, 595, 83-114

Chen, L. & Stone, M. C. Influence of bed material size heterogeneity on bedload transport uncertainty, Water Resources Research, 2008, 44, W01405

Curran, J. C. & Wilcock, P. R. Effect of sand supply on transport rates in a gravel-bed channel J. Hydr. Eng., 2005, 131, 961-967

Furbish, D. J.; Haff, P. K.; Roseberry, J. C. & Schmeeckle, M. W. A probabilistic description of the bed load sediment flux: 1. Theory, Journal of Geophysical Research, 2012, 117, F03031

Turowski, J.M., A. Badoux, D. Rickenmann, 2011, Start and end of bedload transport in gravel bed rivers, Geophysical Research Letters, 38, L04401, doi: 10.1029/2010GL046558

**Anonymous Referee #2**

Review of 'Gravel threshold of motion: A state function of sediment transport disequilibrium'
Earth Surface Dynamics (esurf-2015-52) Joel P. L. Johnson

This paper uses flume experiments and a morphodynamic model to assess the impact that sediment supply has on the evolution of thresholds of motion. The topic of the paper is of interest to readers with some interesting findings that are applicable to the wider discipline. However at present the paper is quite, long, 'dense' and difficult to read in parts meaning that the novelty of the paper is somewhat lost in places. The main comment I feel which needs addressing in this paper is the lack of emphasis on the physical underpinnings of how sediment supply affects the thresholds of motion. Whilst the author makes reference to the bed state conditions in the introduction he does not really follow those through in terms of the implications of his findings. This currently leaves the reader wanting more detail in this regard. There are many papers which talk about the effects of both bed state in terms of structure as well as sand content on entrainment thresholds. I think the latter is particularly important for this paper and the author could look at the following papers as a starting point.

I appreciate the comments, and have worked to address them by reorganizing and expanding on the physical explanations for why thresholds of motion can change over time. This topic is presented in section 1.1, and also 2.1. While the effect of sand on thresholds of motion was addressed in the previous version, it is now discussed much more prominently near the beginning of the manuscript.

Curran, J.C. and Wilcock, P.R. (2005). Effect of sand supply on transport rates in gravel bed channels. Journal of Hydraulic Engineering. 131:961-967

Ikeda, H. and Iseya, F.(1988). Experimental study of heterogeneous sediment transport. Environmental Research Centre Paper 12. University of Tsukuba; Japan.

Jackson, W. L., and Beschta,R.L. (1984). Influences of increased sand delivery on the morphology of sand and gravel channels. Journal of the American Water Resources Association. 20; 527–533.

I now reference these papers in regards to sand supply and thresholds of motion.

I also feel the paper could benefit from being shortened as it is currently quite long and loses focus in places. Detailed comments are also given below.

I have tried hard to improve the paper by following the comments of all of the reviewers. This includes adding material to explain many points further, hopefully making the manuscript less dense. While I have also cut text (in particular Figure 11 and the text along with it), unfortunately the manuscript is now slightly longer as measured by text. However, I also believe that it is more clear and understandable. The revised manuscript better guides the reader and explains why certain things are being presented, hopefully helping it keep its focus (for example, new lines 79-82, 203-211, 386-389, and 545-548).

Line 83- I am not sure I agree with the statement that is still only believed to be controlled by grain parameters. There is an increasing recognition that, as the author alludes, bed state controls are also important. I think at the very least this should be recognised in the current text and references made to the large body of work relating to the impact of structure on bed stability. How does this also link to the concept of mobile armours? You go on to mention this in lines 153-157 so this section could be reorganised?

Section 1.1 now more clearly lays out the relations and overlap between what I am categorizing as grain controls and bed state controls (new lines 84-87). I now specifically address and reference armoring in relation to both of these categories (new lines 92-95). Because space is limited I do not expand on the differences between mobile and static armors.

Line 93- comma missing after vertical position
Added comma.

Lines 123- 131- this section is clumsy and needs re-writing
I edited this section to use active voice and to be less awkward. (new lines 203-211).

Line 141- 143- does this not assume that the bed state does not change? You could have the same overall flux of sediment but the surface structure may change and hence the distribution of threshold stress will thus change as the bed is more stable?
I rearranged this section to have the mention of steady state at the end of the section rather than at the beginning, to more clearly explain how the proposed feedbacks work. I also expanded specifically on the case of steady state threshold of motion based on this comment: Yes, the threshold could evolve under constant flux, but because sediment transport rate and the threshold of motion are directly linked, a change in threshold would change the transport rate (new lines 268-274).

Line 155- consider revision of little additional decrease
Changed wording.

Lines 158 – 167 – I think if you are using the terms interchangeably throughout the paper then there is no need for this paragraph at all.
I got rid of some of this paragraph, and shorted and moved part of it to elsewhere (new lines 52-57). I believe it is important to explicitly state what threshold stresses and reference stresses are, and to justify using them interchangeably.

Line 185- should be dimensional not dimensionally
Changed.

Line 189 – move 'Ar is an optional dimensionless armouring parameter, described further below' to line 198 where you talk about Ar. I think the Ar should be defined as it can have different definitions.

Done; I moved text around and also expanded on this portion, to explain and justify this model parameter (new lines 307-316).

Line 202- this sentence does not make sense- do you mean large grains rather than large range?
That is exactly what I meant; I somehow put the wrong word. I'm glad the reviewer was able to figure out the intention of the sentence.

Line 205- although this paper is concentrating on step pool sequences perhaps something to consider later on in the paper is how applicable these results are to gravel bed rivers more broadly e.g. at lower slopes?
I have added a statement to the discussion that the model parameters were calibrated to these particular steep slopes, and that future work is required to validate the model over a broader range of parameter space. (new lines 635-639).

Line 207 – unit missing after flume length
Done, added m.

Line 226- can you be more specific- how much erosion?
In the interest of clarity and length, I edited the text to focus on the bed responses (coarsening and roughening and sediment transport rates) that matter more for my analysis; in doing so I cut the explicit mention of erosion. (new lines 343-344). Erosion amounts are less informative and somewhat different for the different experiments.

Line 227 – what does 'very low' mean? Can you quantify?
I have reworded the text to say that flux dropped by approximately 3 orders of magnitude, and also have a callout to Figure 2a that shows the transport rate changes through time. (new lines 341-344).

Line 228 – why was this feed rate chosen? What was this rate in comparison to the initial transport rates?
I added an explanation of why the feed rate was chosen—"this feed rate was chosen to be similar to the high initial transport rates (Fig. 2a), while not so high as to inhibit morphodynamic feedbacks by fully burying the stabilized bed surfaces." (new lines 347-349).

Lines 243- consider deleting to GSDs compared
Done

Lines 237- why was the Wilcock and Crowe model specifically used?
I now give the specific reasons that I used this model: because it can account for both surface grain size changes and also let me evaluate whether sand supply can explain the experimental transport trends. (new lines 359-365).

Lines 237- 265- can this section be shortened? Why not just reference the W&CM highlighting the changes you made to it?(lines 262-263)

I considered cutting some of the equations that are Wilcock and Crowe (2003) model, but in the end decided to leave them in. I believe that cutting Eq. 13 and 14 (previously 10 and 11), which show what nondimensional bedload transport rate means and how thresholds of motion actually

go into the transport relation, would make the paper more difficult to understand for most readers, especially those not intimately familiar with W&CM. Also, since I made changes to equations 15 and 16 (previously 12 and 13) I would have to leave those equations in the manuscript; the length of writing actually cut would be pretty small.

Lines 313 -316 – what was your GSD? This is important if you are beginning to duscss sand content and the mechanisms by which sediment feed rate affects initiation of motion? Also in line 313 you mention that the % of grains smaller than 2mm was very small bu tin lines 316 you say 2.8mm was your smallest grain fraction?

I now clarify and describe more completely the full grain size distributions used in the flume experiments; relevant here is that the smallest size class used in the experiments had a D16 of 2.0 mm, D50=2.4 mm, and D84 of 2.8 mm (new lines 327-330). I clarify that this was the size class used for the calculations of sand fraction (new lines 414-422).

Lines 332 – define RMSD
Done (new line 433)

Lines 336 – 344 – this is an interesting finding but what are the implications of this in terms of bed state?
I now give a suggestion of why my model was seemingly insensitive to having combinations of D84, D50, D16 and bed roughness included as another parameter in the model—it may suggest that net erosion and deposition were more important over the range of parameter space explored in the experiments (new lines 440-442).

Line 352 – need full reference to Parker
Done, new line 451.

Line 473- change stresses to stress
Done

Line 474- I am not sure they are comparable are they? Again thinking in terms of the relative effects of bed structure and implications of grain size, structure and thresholds of motion would D50 and D84 be expected to behave the same?
I clarified the writing; I was not trying to suggest that D50 and D84 thresholds would necessarily be or should be expected to be interchangeable. I simply make the point that Recking's relation is not too far off from my thresholds of motion, although the R^2 value is still low (new lines 552-560).

Line 475- what do you mean by 'fairly comparable'?
Reworded to say I do not expect these different threshold measured to be equivalent (new lines 553-554).

Lines 503-506- I think this is one of the places where a better physical explanation behind the findings would be useful
Good point. I added a substantial amount of text. First, I now acknowledge that future work would be required to really determine specific process linkages explaining asymmetry in aggradation vs degradation effects on thresholds of motion (new lines 587-589). Second, I present a hypothesis that could be tested with future work about the differences in deposition, erosion and roughness evolution (new lines 589-606).

Lines 530 – this section is supposed to be linked to system memory but I find it hard to distinguish this and a much more explicit link needs to be made.

While the section talked about system memory, it also covered other topics.  In the interest of article length I removed the section (old section 4.3 in the previous manuscript, and also figure 11) and also cut most of the content.  I did however keep and expand slightly on parts related to memory, these are now at new lines 628-639.

Lines 545-546 – I would re-write to avoid asking a rhetorical question
This sentence was cut during editing.

Lines 576 – whilst I find this section an interesting concept I think it could be shortened a lot given the paper is already quite long.

I shortened this section by roughly 23% through editing (new section 4.2).  However, I feel like it is an important idea to explain thoroughly, and I really do not want to remove more of it. I did cut substantial portions of old section 4.3 in the previous manuscript, and also old figure 11, in order to keep the manuscript focused and not even longer.

Line 584- expand upon the work of Phillips (2007)
Done, new lines 649-652.

Line 625 - I would re-write to avoid asking a rhetorical question

Done.  I also edited out some other rhetorical questions in the manuscript.

**Anonymous Referee #3**

I believe this is an overall excellent piece of work, written by an expert in the field. The issue of sediment transport is a long studied problem and much attention has been paid to traditional criteria, such as Shield's critical shear stresses (as the author notes himself). There are a number of problems using such criteria - as the author mentions in his work (also demonstrated in Fig. 1). However, the author still chooses to deploy this criterion focusing on the fact that data scatter (e.g. in Fig.1) is due to a range of factors, however omitting to discuss its inability to represent the rich dynamics of grain transport, as recent research has shown (Schemeeckle et al. 2003, Diplas et al. 2008).

The major novelty of the present work lays in the presentation of a state function for the description of sediment transport, which is a very much welcomed contribution as a conceptual approach. However, there is a significant concern (to this reviewer) over the suitability of the Shield's shear stress as parameter to be used in this model. Would not other more criteria that capture the full range of grain dynamics, such as instantaneous hydrodynamic forces near the bed or even better the impulse/energy content of flow structures, be more suitable as model parameters? Of course such analysis may offer enough new material for another (and perhaps more impactful) publication, but yet it may be useful to add a note about this on the discussion section.

I appreciate the review and the different perspective it provides.  I now cite the work by Schmeeckle and Diplas, and also statistical mechanics descriptions of bedload transport by Furbish et al at the start of section 2.1 (new lines 219-220).  In lines 214-221 I also address the reviewer's comment in another way, by more specifically defining the narrower "parameter

space" of the model, and the limits of the model. I explicitly state that the model intentionally does not describe timescales of turbulent velocity fluctuations, and I also state that the model is deterministic rather than stochastic. I agree that there are rich bedload transport dynamics over timescales of turbulent velocity fluctuations. I also believe that my model is new and novel in its ability to explore rich morphodynamic feedbacks that have not yet been modeled well, over timescales longer than turbulence.

Another, minor issue is with the interpretation of the data analysis. In particular, is there no better measure to assess the "amount of information embedded" between two variables than R2? R2 is rather demonstrative of the strength of association between two variables.

I also use RMSD (and define it) in the manuscript. While I am interested in finding new and better statistical tests that could do more than determine the strength of corrrelations between variables, I do not know what other statistics would actually be better for my applications, and I respectfully note that the reviewer does not provide any specific suggestion for statistical tests to include either.

---

## Author Comment (AC4) · 31 Jul 2016

I believe this is an overall excellent piece of work, written by an expert in the field. The issue of sediment transport is a long studied problem and much attention has been paid to traditional criteria, such as Shield's critical shear stresses (as the author notes himself). There are a number of problems using such criteria - as the author mentions in his work (also demonstrated in Fig. 1). However, the author still chooses to deploy this criterion focusing on the fact that data scatter (e.g. in Fig.1) is due to a range of factors, however omitting to discuss its inability to represent the rich dynamics of grain transport, as recent research has shown (Schemeeckle et al. 2003, Diplas et al. 2008).

The major novelty of the present work lays in the presentation of a state function for the description of sediment transport, which is a very much welcomed contribution as a conceptual approach. However, there is a significant concern (to this reviewer) over the suitability of the Shield's shear stress as parameter to be used in this model. Would not other more criteria that capture the full range of grain dynamics, such as instantaneous hydrodynamic forces near the bed or even better the impulse/energy content of flow structures, be more suitable as model parameters? Of course such analysis may offer enough new material for another (and perhaps more impactful) publication, but yet it may be useful to add a note about this on the discussion section.

I appreciate the review and the slightly different perspective it provides. I now cite the work by Schmeeckle and Diplas, and also statistical mechanics descriptions of bedload transport by Furbish et al at the start of section 2.1 (new lines 219-220). In lines 214-221 I also address the reviewer's comment in another way, by more specifically defining the narrower "parameter space" of the model, and the limits of the model. I explicitly state that the model intentionally does not describe timescales of turbulent velocity fluctuations, and I also state that the model is deterministic rather than stochastic. I agree that there are rich bedload transport dynamics over timescales of turbulent velocity fluctuations. I also believe that my model is new and novel in its ability to explore rich morphodynamic feedbacks that have not yet been modeled well, over timescales longer than turbulence.

Because of how Word changes line numbers in the "track changes" version of the manuscript, I note that the line numbers refer to the revised manuscript that does not show all of the edits.

Another, minor issue is with the interpretation of the data analysis. In particular, is there no better measure to assess the "amount of information embedded" between two variables than R2? R2 is rather demonstrative of the strength of association between two variables.

In addition to $R^2$, I also use RMSD (and define it) in the manuscript. While I am interested in finding new and better statistical tests that could do more than determine the strength of corrrelations between variables, I do not know what other statistics would actually be better for my particular applications, and I respectfully note that the reviewer does not provide any specific suggestion for statistical tests to include either.